

# 1     Is water vapor a key player of the wintertime haze in North China Plain?

Jiarui Wu[1,4,7], Naifang Bei[2], Bo Hu[3], Suixin Liu[1,4], Meng Zhou[5], Qiyuan Wang[1,4], Xia Li[1,4,7], Lang Liu[1,4,7], Tian
Feng[1], Zirui Liu[3], Yichen Wang[1], Junji Cao[1,4], Xuexi Tie[1,4], Jun Wang[5], Luisa T. Molina[6], and Guohui Li[1,4*]
[1]Key Lab of Aerosol Chemistry and Physics, SKLLQG, Institute of Earth Environment, Chinese Academy of
Sciences, Xi'an, Shaanxi, China
[2]School of Human Settlements and Civil Engineering, Xi'an Jiaotong University, Xi'an, Shaanxi, China
[3]State Key Laboratory of Atmospheric Boundary Layer Physics and Atmospheric Chemistry, Institute of
Atmospheric Physics, Chinese Academy of Sciences, Beijing, 100029, China
[4]CAS Center for Excellence in Quaternary Science and Global Change, Xi'an, China
[5]Department of Chemical and Biochemical Engineering & Interdisciplinary Graduate Program in
Geo-Informatics, University of Iowa, Iowa City, Iowa, USA
[6]Molina Center for Energy and the Environment, La Jolla, California, USA
[7]University of Chinese Academy of Science, Beijing, China
[*]Correspondence to: Guohui Li (ligh@ieecas.cn)
**Abstract.** Water vapor has been proposed to amplify the severe haze pollution in China by
enhancing the aerosol-radiation feedback (ARF). Observations have revealed that the
near-surface $PM_{2.5}$ concentrations ($[PM_{2.5}]$) generally exhibits an increasing trend with
relative humidity (RH) in North China Plain (NCP) during 2015 wintertime, indicating that
the aerosol liquid water (ALW) caused by hygroscopic growth could play an important role in
the $PM_{2.5}$ formation and accumulation. Simulations during a persistent and heavy haze
pollution episode from 05 December 2015 to 04 January 2016 in NCP were conducted using
the WRF-CHEM model to comprehensively quantify contributions of the ALW effect to
near-surface $[PM_{2.5}]$. The WRF-CHEM model generally performs reasonably well in
simulating the temporal variations of RH against measurements in NCP. The factor separation
approach (FSA) was used to evaluate the contribution of the ALW effect on the ARF,
photochemistry, and heterogeneous reactions to $[PM_{2.5}]$. The ALW not only augments particle
sizes to enhance aerosol backward scattering, but also increases the effective radius to favor
aerosol forward scattering. The contribution of the ALW effect on the ARF and
photochemistry to near-surface $[PM_{2.5}]$ is not significant, generally within 1.0 μg m$^{-3}$ on
average in NCP during the episode. Serving as an excellent substrate for heterogeneous
reactions, the ALW substantially enhances the secondary aerosol (SA) formation, with an
average contribution of 71%, 10%, 26%, and 48% to near-surface sulfate, nitrate, ammonium,
and secondary organic aerosol concentrations. Nevertheless, the SA enhancement due to the
ALW decreases the aerosol optical depth and increases the effective radius to weaken the
ARF, reducing near-surface primary aerosols. The contribution of the ALW total effect to
near-surface $[PM_{2.5}]$ is 17.5% on average, which is overwhelmingly dominated by enhanced
SA. Model sensitivities also show that when the RH is less than 80%, the ALW progressively



increases near-surface [PM$_{2.5}$], but commences to decrease when the RH exceeding 80% due
to the high occurrence frequencies of precipitation.





## 1 Introduction


Atmospheric aerosols or fine particle matters (PM$_{2.5}$) influence the climate directly by
scattering and absorbing the solar radiation, and indirectly by serving as cloud condensation
nuclei and ice nuclei (Ackerman, 1977; Ackerman and Baker, 1977; Jacobson, 1998, 2002;
Penner et al., 2001). Moreover, high levels of PM$_{2.5}$ in the atmosphere also cause severe haze
pollution, impairing visibility and exerting deleterious effect on ecological system and human
health (Chan and Yao, 2008; Zhang et al., 2013; Kurokawa et al., 2013; Weinhold, 2008;
Parrish and Zhu, 2009). In addition to anthropogenic emissions, the poor air quality is
generally influenced by stagnant meteorological situations with weak winds and high relative
humidity (RH) (Flocas et al., 2009; Quan et al., 2013; Zhang et al., 2014; Bei et al., 2016; Wu
et al., 2017; Ding et al., 2017). RH, as an important meteorological factor in the atmosphere,
considerably affects the formation, chemical composition, and physical properties of
atmospheric aerosols (Seinfeld et al., 2001; Hallquist et al., 2009; Poulain, 2010; Nguyen et
al., 2011).
As the main constituent in the atmosphere, water vapor directly participates in the
atmospheric physical and chemical processes. Since many components of atmospheric
aerosols are hygroscopic, they take up water as RH increases (Covert et al., 1972; Pilinis et
al., 1989), thereby influencing the aerosol size distribution, chemical composition, mass
concentration, and corresponding optical properties as well as radiative effects (Im et al.,
2001; Carrico et al., 2003; Randles et al. 2004; Cheng et al., 2008). Wang et al. (2016) have
indicated that the ratio of SO$_4^{2-}$ to SO$_2$ exhibits an exponential increase with RH. Tie et al.
(2017) have shown that the sulfate, nitrate, and ammonium concentrations increase from 16
to 25 µg m$^{-3}$, 15 to 23 µg m$^{-3}$, and 11 to 17 µg m$^{-3}$, respectively, when RH increases from 60%
to 80%. Field measurements in Beijing have demonstrated that the inorganic aerosol fraction
increases with increasing RH (Wu et al., 2018). In addition, water vapor also serves as an



important medium in the formation of secondary aerosols (SA) through liquid-phase
reactions and heterogeneous reactions (Seinfeld and Pandis, 1986; Pilinis et al., 1989). For
example, Li et al. (2017) have indicated that the aerosol liquid water (ALW) induced by the
wet growth could play a significant role in the sulfate formation and emphasized the
importance of bulk aqueous-phase oxidation of $SO_2$ in ALW and heterogeneous reaction of
$SO_2$ on aerosol surfaces involving ALW. ALW also plays an important role in secondary
organic aerosol (SOA) formation (Hastings et al., 2005; Healy et al., 2009; Kamens et al.,
2011; Koehler et al., 2004). Numerous studies have investigated the effect of RH on SOA
formed from different aromatics during their photochemical oxidation processes (Blando and
Turpin, 2000; Cocker et al., 2001; Seinfeld et al., 2001; Zhou et al., 2011; Jia and Xu., 2014).
Furthermore, Zhang et al. (2015) have revealed that, as the RH increases from 40% to 85% in
the Yangtze River Delta of China, the aerosol scattering and backscattering coefficients
increase by 58% and 25%, respectively, and the calculated aerosol direct radiative forcing
caused by hygroscopic growth is increased by 47%.

In recent years, China has experienced persistent haze pollution with unprecedentedly

high $PM_{2.5}$ concentrations during wintertime, particularly in North China Plain (NCP) (Chan
and Yao, 2008; He et al., 2001; Kan et al., 2012; Guo et al., 2014; Wang et al., 2014; Fu et al.,
2014). A conceptual model based on the aerosol radiation feedback (ARF) has been
established to interpret the wintertime heavy haze formation, in which water vapor is
considered to play a key role in the progressive accumulation and formation of $PM_{2.5}$. In
winter, when the atmospheric condition is stagnant, air pollutants commence to accumulate in
the planetary boundary layer (PBL), favorable for the $PM_{2.5}$ formation. Increasing $PM_{2.5}$
scatters or absorbs the incoming solar radiation to lower the surface temperature and cause
anomalous temperature inversion, subsequently suppressing the vertical turbulent diffusion
and decreasing the planetary boundary layer height (PBLH) to further trap more air pollutants



and water vapor to increase the RH in the PBL. Increasing RH enhances aerosol hygroscopic
growth and multiphase reactions and augments the particle size and mass, causing further
dimming and decrease of the surface temperature and PBL height (Quan et al., 2013; Tie et
al., 2017; Ding et al., 2017). However, few studies have been performed in China to
comprehensively quantify the effect of water vapor in the atmospheric physical and chemical
process on the $PM_{2.5}$ pollution to further verify the haze formation.

The purpose of the present study is to quantitatively evaluate the contribution of aerosol

water induced by the aerosol wet growth to $PM_{2.5}$ concentrations in NCP using the Weather
Research and Forecast model with Chemistry (WRF-CHEM). The model configuration and
methodology are described in Section 2. Results and discussions are presented in Section 3,
and conclusions and summaries are given in Section 4.

**2    Model and methodology**
**2.1  WRF-CHEM Model and Configuration**

A persistent air pollution episode with high levels of $PM_{2.5}$ from 05 December 2015 to

04 January 2016 in NCP was simulated using the WRF-CHEM model with modifications by
Li et al. (2010, 2011a, b, 2012) from the Molina Center for Energy and the Environment.
Figure 1 shows the WRF-CHEM model simulation domain and Table 1 provides the model
configurations. Detailed model description can be found in Wu et al. (2018a).
**2.2  Data and Methodology**

The model performance of RH was validated using the hourly measurements in

Luancheng, Yucheng, and Jiaozhouwan observed from the Chinese Ecosystem Research
Network (CERN). Furthermore, the NCEP reanalysis data was used to compare to the
simulated RH distribution. The detailed information of other data used for validation can be
found in Wu et al. (2018a).



The mean bias (*MB*), root mean square error (*RMSE*) and the index of agreement (*IOA*)
were utilized to evaluate the performance of the WRF-CHEM model simulations against
measurements. To assess the contributions of ALW to the near-surface concentrations of air
pollutants in NCP, the factor separation approach (FSA) was used in this study (Stein and
Alpert, 1993; Gabusi et al., 2008; Li et al., 2014). Generally, the formation of the secondary
atmospheric pollutants, such as $O_3$, secondary organic aerosol, and nitrate, is a complicated
nonlinear process in which its precursors from various emissions sources and transport react
chemically or reach equilibrium thermodynamically. Nevertheless, it is not straightforward to
evaluate the contributions from different factors in a nonlinear process (Wu et al., 2017). The
factor separation approach (FSA) proposed by Stein and Alpert (1993) can be used to isolate
the effect of one single factor from a nonlinear process and has been widely used to evaluate
source effects. The total effect of one factor in the presence of others can be decomposed into
contributions from the factor and that from the interactions of all those factors. Considering
that there are two factors *X* and *Y* that influence the formation of secondary pollutants in the
atmosphere and also interact with each other. Denoting $f_{XY}$, $f_X$, $f_Y$, and $f_0$ as the
simulations including both of two factors, factor *X* only, factor *Y* only, and none of the two
factors, respectively. The contributions of factor *X* and *Y* can be defined as $f_{XY} - f_Y$ and
$f_{XY} - f_X$, respectively. Detailed description of the methodology can be found in Wu et al.

(2017).


**3    Results and discussions**
**3.1  Relationship between RH and near-surface PM$_{2.5}$ concentrations**
High RH has been suggested to be an important factor facilitating the SA formation in
the atmosphere and aggravating the haze pollution (Sun et al., 2013; Cheng et al., 2015).
Figure 2 presents the scatter plot of the near-surface PM$_{2.5}$ concentrations ([PM$_{2.5}$]) and RH in





the winter of 2015 at six typical polluted cities in NCP, including Beijing, Tianjin,
Shijiazhuang, Tangshan, Baoding, and Chengde. The observed near-surface [$PM_{2.5}$] at those
six cities display a growing trend with increasing RH, suggesting that the ALW induced by
the hygroscopic growth under high RH conditions has potentials to accelerate the $PM_{2.5}$
formation and accumulation. Increasing RH facilitates the aerosol hygroscopic growth and
further enhances the ALW, which serves as an efficient medium for promoting the
liquid-phase and heterogeneous reactions and accelerating the transformation of reactive
gaseous pollutants into the particle phase. Increased ALW also augments the particle size,
enhancing the ARF to increase the near-surface [$PM_{2.5}$]. However, the attenuation of
incoming solar radiation caused by the ALW also decreases the photolysis rates, unfavorable
for photochemical activities and lowering the atmospheric oxidation capability (AOC). Field
measurements show that large fraction of SA in $PM_{2.5}$ has been observed in NCP during
wintertime (Sun et al., 2013; Guo et al., 2014; Xu et al., 2015). Therefore, decreased AOC
generally does not facilitate the SA formation, particularly with regards to SOA and nitrate, to
partially counteract the $PM_{2.5}$ enhancement caused by the ALW. It is also worth noting that
since high RH frequently corresponds to atmospheric stagnation, near-surface [$PM_{2.5}$] also
build up under high RH conditions. For example, the humid air mass is subject to being
transported from south to NCP under the stagnant weather with weak south winds and
meanwhile the $PM_{2.5}$ also accumulates due to the unfavorable dispersion condition.
Additionally, when the RH is very high, there also exist the low near-surface [$PM_{2.5}$] shown
in Figure 2, demonstrating that other factors, such as emissions, horizontal transport, vertical
exchange, and precipitation, also substantially influence near-surface [$PM_{2.5}$]. Generally, high
occurrence frequency of precipitation coincides with high RH, thus the precipitation washout
might constitute one of the most possible reasons for the low near-surface [$PM_{2.5}$] under high
RH situations. Therefore, it is still imperative to verify quantitatively the contribution of the





ALW to near-surface [$PM_{2.5}$].
**3.2  Model validation**

The WRF-CHEM model simulation of the haze pollution episode in NCP has been

comprehensively validated using available measurements in Wu et al. (2018a). In general, the
model simulates well the spatial distribution and temporal variation of $PM_{2.5}$, $O_3$, $NO_2$, $SO_2$
and CO mass concentrations compared to observations in NCP. The predicted aerosol species
are also in good agreement with the measurement in Beijing. Moreover, the model performs
reasonably well in simulating the aerosol optical depth and single scattering albedo, PBL
height and downward shortwave flux against measurements.

In order to verify the effect of the ALW on near-surface [$PM_{2.5}$] during the haze

pollution episode, the simulated temporal variation of RH was first compared to
measurements at Luancheng, Yucheng, and Jiaozhouwan in NCP from 05 December 2015 to
04 January 2016 (Figure 3). The WRF-CHEM model generally performs well in simulating
the hourly variation of RH in these three cities, with *IOA*s of 0.73, 0.83, and 0.69,
respectively. RH is a key meteorological component, sensitive to the atmospheric
thermodynamic (e.g., temperature) and dynamic (e.g., winds) conditions. Even when the
simulated water vapor content is the same as the observation, the overestimation or
underestimation of temperature still causes underestimation or overestimation of RH. Biases
of wind speeds and directions considerably influence the origination of air mass at the
observation site. In general, the northerly wind carries dry air, and is opposite for the
southerly wind during wintertime in NCP. Therefore, the uncertainties from meteorological
field simulations might constitute one of the most possible reasons for the RH bias (Bei et al.,
2017). Figure 4 presents the pattern comparison of the average simulated RH and the NCEP
reanalysis during the episode. The simulated RH distribution is generally consistent with that
from the reanalysis, e.g., dry air in West China and fairly humid air in South China. However,



the air over NCP in the simulation is more humid than the analysis, and the average simulated
RH is 70.6%, about 20% higher than the reanalyzed RH, which might be caused by the
temperature decrease due to the ARF considered in the simulation.
**3.3 Sensitivity studies**
The ALW not only enlarges the particle size to increase the aerosol optical depth (AOD),
likely enhancing the ARF to facilitate the $PM_{2.5}$ accumulation or to alter photolysis rates to
affect the AOC, but also influences the SA formation serving as a medium for multiphase
reactions. Therefore, sensitivity studies are used to quantitatively evaluate the effect of the
ALW on the $PM_{2.5}$ concentration during the haze pollution episode.
The FSA method was used to evaluate the contribution of the ALW to near-surface
$[PM_{2.5}]$ by differentiating two model simulations with and without the ALW effect. Besides
the base case with all the ALW effect (hereafter referred as to $f_{base}$), additional four
sensitivity simulations were performed, in which the ALW effect on the ARF, photolysis,
multiphase reactions, and the total were excluded, respectively (hereafter referred as to
$f_{alw-rad0}$, $f_{alw-j0}$, $f_{alw-het0}$, and $f_{alw-tot0}$, respectively)
**3.3.1 ALW effect on the ARF**
The ALW, caused by the aerosol hygroscopic growth, augments the particle size to
increase AOD, potentially enhancing the ARF and aggravating the haze pollution. Figure 5
shows the distribution of the average AOD contribution due to the ALW during the haze
episode, evaluated by differentiating $f_{base}$ and $f_{alw-rad0}$. Apparently, the ALW
substantially increases the AOD in NCP, with the contribution ranging from 30% to more
than 50%, indicating that ALW is an important contributor of the AOD. Substantial increase
of the AOD due to the ALW is anticipated to attenuate the incoming solar radiation,
decreasing the surface temperature and suppressing the PBL development, therefore,
deteriorating the haze pollution, as proposed by recent studies (Tie et al., 2017; Liu et al.,



2018).

Figure 6 presents the distribution of the average near-surface $PM_{2.5}$ contribution of the
ALW effect on the ARF (hereafter referred as to ALW-ARF) during the haze episode.
Interestingly, the ALW-ARF does not increase the near-surface $[PM_{2.5}]$ consistently in NCP,
as expected. The ALW-ARF enhances the near-surface $[PM_{2.5}]$ most strikingly in the south of
Hebei, with a contribution of more than 15 μg m$^{-3}$ (less than 7%). However, in some areas,
such as the east of Shandong, the near-surface $PM_{2.5}$ contribution of the ALW-ARF is
negative, or the ALW-ARF decreases $[PM_{2.5}]$. On average, the ALW-ARF increases
near-surface $[PM_{2.5}]$ in NCP during the episode by about 1.1 μg m$^{-3}$, so cannot constitute an
important factor for the heavy haze formation.
It is worth noting that enlarged particles due to the ALW not only increase the AOD to
enhance aerosol backward scattering, but also cause the aerosol spectrum to successively
shift toward larger sizes. Based on the Mie scattering theory, when the particle size is similar
to the wavelength of incoming solar radiation, the radiation is favored to be scattered in the
forward directions. In order to further verify the ALW effect on solar radiation and
near-surface $[PM_{2.5}]$, an ensemble method is employed similar to that reported in Wu et al.
(2018a). The daytime near-surface $[PM_{2.5}]$ in NCP during the episode in $f_{base}$ are first
subdivided into 30 bins with the interval of 20 μg m$^{-3}$. The AOD at 550 nm, aerosol effective
radius (Reff), downward shortwave radiation at the surface (SWDOWN), surface temperature
(TSFC), PBLH, and near-surface $[PM_{2.5}]$ in $f_{base}$ and $f_{alw-rad0}$ in the same grid cell are
assembled as the bin $[PM_{2.5}]$, respectively, and an average of these variables in each bin are
calculated. Figure 7 shows the variation of AOD and Reff in $f_{base}$ and $f_{alw-rad0}$ as a
function of bin $[PM_{2.5}]$. The ALW not only significantly enhances the AOD, with an average
contribution of 46% in NCP during the episode, but also increases the Reff considerably
(Figures 7a and 7b). The Reff enhancement due to the ALW is the most striking with



near-surface [PM$_{2.5}$] between 40 and 160 μg m$^{-3}$, exceeding 60%. On average, the ALW
increases the Reff from 0.31 μm to 0.48 μm, close to the peak band of solar radiation.
Therefore, the ALW increases the AOD to scatter more incoming solar radiation, decreasing
the SWDOWN, but augments particle sizes to favor the forward scattering, increasing the
SWDOWN. Generally, the decrease of the SWDOWN caused by the ALW is not significant,
less than 1% with near-surface [PM$_{2.5}$] less than 200 μg m$^{-3}$ and ranging from 2% to 3% with
[PM$_{2.5}$] exceeding 240 μg m$^{-3}$ (Figure 8a). Correspondingly, the ALW-ARF effect on the
daytime TSFC is also marginal and the TSFC is decreased by less than 0.2$^{o}$C (Figure 8b).
Furthermore, the ALW-ARF generally increases the daytime PBLH with near-surface [PM$_{2.5}$]
less than 140 μg m$^{-3}$, but are opposite with [PM$_{2.5}$] exceeding 140 μg m$^{-3}$ (Figure 8c). Hence,
the contribution of the ALW-ARF to near-surface [PM$_{2.5}$] is highly uncertain (Figure 8d),
depending on the relative importance of the ALW induced enhancement of aerosol backward
and forward scattering.
**3.3.2 ALW effect on the photochemistry**
In addition to the ARF, the ALW also exerts an impact on the photochemistry by altering
the aerosol backward and forward scattering to affect the photolysis, further influencing the
ozone (O$_3$) and SA formation. Previous studies have shown that the ALW modifies the
vertical profile of photolysis rate of NO$_2$ ($J_{NO_2}$), inhibiting it at the ground level and
accelerating it in the upper PBL (Tao et al., 2014; Dickerson et al., 1997). The combination
reaction between the ground state oxygen atom (O$^3$P), produced from NO$_2$ photolysis, and
oxygen molecules (O$_2$) forms O$_3$, representing the only important source of O$_3$ in the
troposphere:
$$NO_2 + h\nu \rightarrow NO + O(^3P) \tag{1}$$
$$O(^3P) + O_2 + M \rightarrow O_3 + M \tag{2}$$
Thus, the variation of $J_{NO_2}$ considerably affects the O$_3$ formation in the troposphere,



changing the AOC and further the SA formation.
Figures 9a and 9b present the distribution of the percentage variation of average daytime
$J_{NO_2}$ due to the ALW during the haze episode at the $1^{st}$ and $5^{th}$ model layer, respectively. At
the $1^{st}$ model layer, except in the north of Jiangsu, the ALW generally decreases $J_{NO_2}$
slightly in NCP. However, at $5^{th}$ model layer, the region with enhanced $J_{NO_2}$ caused by the
ALW is obviously increased compared to that at $1^{st}$ model layer. Apparently, the ALW
induced enlargement of particle sizes increases the AOD and enhances the aerosol backward
scattering to reduce the solar radiation reaching the ground level, decreasing the photolysis
rate, but the enhanced forward scattering still potentially accelerates the photolysis, such as in
the north of Jiangsu. In addition, the enhanced aerosol backward scattering also increases the
photolysis rate in the upper and above PBL, which is consistent with previous studies (Tao et
al., 2014; Dickerson et al., 1997). The variation of average daytime $O_3$ concentrations is not
consistent with that of $J_{NO_2}$ at the $1^{st}$ model layer (Figure 9c). For example, although the
$J_{NO_2}$ is decreased in Hebei and Shandong, the $O_3$ concentration is still enhanced by the ALW
in some areas of the two provinces. One of the possible reasons is the vertical transport of $O_3$
from the upper layers where the $O_3$ is plausibly enhanced due to the increased photolysis rate
(Figure 9b). At the $5^{th}$ model layer, the area with enhanced $O_3$ concentrations is much larger
than that at $1^{st}$ model layer, which is in agreement with the variation of the $J_{NO_2}$ (Figure 9d).
On average, the ALW decreases near-surface ($1^{st}$ model layer) daytime $O_3$ concentrations by
about 0.2 µg m$^{-3}$ (or 0.45%), playing a minor role in $O_3$ formation.
Figure 10 shows the distribution of the average near-surface $PM_{2.5}$ contribution of the
ALW effect on the photolysis frequencies (hereafter referred as to ALW-J) during the haze
episode. Except in some areas in Shandong and Anhui, the ALW-J slightly decreases
near-surface [$PM_{2.5}$] in NCP, with an average reduction of about 0.87 µg m$^{-3}$ (or 0.64%).
Therefore, the ALW-J does not play an important role in mitigating the haze pollution.



### 3.3.3 ALW effect on heterogeneous reactions

The ALW provides an excellent substrate for heterogeneous reactions in the atmosphere, which have been proposed to play a key role in the SA formation during haze days (Li et al., 2017; Xing et al., 2018).

Figure 11 presents the distribution of contributions of the ALW on heterogeneous reactions (hereafter referred to ALW-HET) to near-surface sulfate, nitrate, and ammonium concentrations averaged during the haze episode. A parameterization of sulfate heterogeneous formation involving ALW has been developed and implemented into the WRF-CHEM model, which has successfully reproduced the observed rapid sulfate formation during haze days (Li et al., 2017). The sulfate heterogeneous formation from $SO_2$ is parameterized as a first order irreversible uptake by ALW surfaces, with a reactive uptake coefficient of $0.5\times10^{-4}$ assuming that there is enough alkalinity to maintain the high iron-catalyzed reaction rate. The contribution of the ALW-HET to the sulfate formation is substantial in NCP, exceeding 5 μg $m^{-3}$ (or 50%) in NCP (Figures 11a and 11b). The ALW-HET contributes about 71.3% of the sulfate in NCP during the episode, indicating that the heterogeneous formation involving the ALW is the dominant sulfate source during haze days.

The heterogeneous hydrolysis of $N_2O_5$ on the surface of deliquescent aerosols leads to the $HNO_3$ formation, which is the key contributor to the nitrate aerosol loading (Riemer et al., 2003; Pathak et al., 2011; Chang et al., 2016). In this study, the parameterization of the heterogeneous hydrolysis of $N_2O_5$, proposed by Riemer et al. (2003), has been included in the WRF-CHEM model for considering the effect of ALW-HET on the nitrate formation. The $N_2O_5$ uptake coefficient on wet aerosol surfaces ranges from 0.002 to 0.02, depending on the sulfate and nitrate aerosol mass. The ALW-HET generally increases the near-surface nitrate mass concentration by 2 to 8 μg $m^{-3}$ in NCP, with an average contribution of 2.8 μg $m^{-3}$ (or 10%) (Figures 11c and 11d). It is worth noting that the ALW-HET does not consistently





increase the nitrate formation and the nitrate concentration is considerably decreased in some
areas in east China. One of the possible reasons is that the ALW-HET substantially enhances
the sulfate formation and the increased sulfate competes with nitrate for ammonia ($NH_3$),
suppressing the nitrate formation. Figure 12 shows the distribution of the $NH_3$ emission rate
in December. High $NH_3$ emissions are concentrated in NCP, Central China, Sichuan basin,
and Northeast China, where the nitrate concentration is generally increased by the ALW-HET.
Ammonium serves as the main alkali in the atmosphere to neutralize acidic aerosols
such as sulfate and nitrate, so its formation is not only dependent on its precursor ($NH_3$), but
is also influenced by acid aerosols. The ALW-HET enhances the near-surface ammonium
mass concentration most strikingly in NCP with high $NH_3$ emissions and increased sulfate
and nitrate aerosols. The contribution of the ALW-HET to the ammonium concentration
varies from 2 to more than 10 µg m$^{-3}$, with an average of 4.2 µg m$^{-3}$ (or 25.6%), showing that
the heterogeneous formation constitutes an important ammonium source (Figures 11e and
11f).
Heterogeneous reactions are also an important pathway for the SOA formation (Fu et al.,
2008; Li et al., 2013). Laboratory and field studies have indicated that glyoxal and
methylglyoxal play an important role in SOA formation via aerosol uptake or cloud
processing (Liggio et al., 2005; Volkamer et al., 2007; Li et al., 2011). In this study, the
heterogeneous reaction of SOA formation from glyoxal and methylglyoxal is parameterized
as a first-order irreversible uptake by aerosol particles, with a reaction uptake coefficient of
3.7 ×10$^{-3}$ (Liggio et al., 2005; Zhao et al., 2006; Volkamer et al., 2007; Li et al., 2011).
During the haze episode, the average near-surface SOA contribution of the ALW-HET is 7.4
µg m$^{-3}$ (or 48%) in NCP, ranging from 3 to over 15 µg m$^{-3}$ (or 30 to over 50%) in NCP,
showing that heterogeneous reactions of glyoxal and methylglyoxal on wet aerosol surfaces
play a critical role in SOA formation (Figure 13). Xing et al. (2018) have demonstrated that,



in Beijing-Tianjin-Hebei, the near-surface SOA contribution from glyoxal and methylglyoxal
is around 30% during a haze episode in January 2014, much less than that (48%) in the study.
The possible reason is that the RH in the episode is higher than that in Xing et al. (2018),
causing more SOA formation from glyoxal and methylglyoxal.

High SA contribution to the haze pollution has been observed in China (e.g., Huang et

al., 2014), so the ALW-HET substantially enhances the SA formation, constituting an
important factor for heavy haze formation. Figure 14 shows the contribution of the
ALW-HET to average near-surface [$PM_{2.5}$] during the study episode. The ALW-HET
enhances near-surface [$PM_{2.5}$] by more than 40 μg m$^{-3}$ in the central part of NCP, and on
average, the $PM_{2.5}$ contribution of the ALW-HET is 21.7 μg m$^{-3}$ (or 15.9%), less than the total
enhancement of 25.1 μg m$^{-3}$ (or 18.4%) from sulfate, nitrate, ammonium, and SOA. The
inconsistency indicates that the ALW-HET induced SA enhancement causes a decrease in
primary aerosols. Figure 15a and 15b show the percentage decrease of black carbon (BC) and
primary organic aerosols (POA) as a function of bin [$PM_{2.5}$], respectively. Interestingly, the
average BC and POA concentrations are decreased by 5.1% and 5.2% due to the ALW-HET,
respectively. The ALW-HET induced SA enhancement augment the particle size and should
increase the AOD. However, on the contrary, the ALW-HET decreases the AOD, although
considerably increases the Reff (Figures 16a and 16b). The reason is that the enhanced SA
enlarges particle sizes, facilitating the coagulation to decrease the aerosol number
concentration and surface area, as shown in Figures 16c and 16d. Hence, the decreased AOD
and increased Reff due to the ALW-HET enhance the SWDOWN and TSFC, further
increasing the PBLH and decreasing near-surface primary aerosols, as shown in Figure 17.
**3.3.4 ALW total effect**

Above discussions have shown that the ALW influences near-surface [$PM_{2.5}$] through

complicated physical and chemical processes, which interact with each other. Figure 18





shows the near-surface $PM_{2.5}$ contribution of the total ALW effect (hereafter referred as to
ALW-TOT) during the haze episode, evaluated by differentiating $f_{base}$ and $f_{alw-tot0}$. The
ALW-TOT plays an important role in the $PM_{2.5}$ formation during the wintertime haze
pollution in NCP, with an average contribution of 23.8 µg m$^{-3}$ (or 17.5%), ranging from 5 to
over 40 µg m$^{-3}$. About 78% of the enhanced near-surface [$PM_{2.5}$] due to the ALW-TOT is
contributed by secondary inorganic aerosols, in which the contributions from sulfate, nitrate,
and ammonium are 45%, 14% and 19%, respectively, and around 32% is contributed by SOA.
Therefore, about 10% of near-surface [$PM_{2.5}$] enhancement from SA is counterbalanced by
the decrease in the primary aerosols. Therefore, the ALW induced enhancement of
near-surface [$PM_{2.5}$] is overwhelmingly determined by the ALW-HET, and the ALW-ARF
and ALW-J are subject to decreasing [$PM_{2.5}$].

Figure 19 shows the variation of near-surface [$PM_{2.5}$] caused by the ALW-TOT,

ALW-HET, ALW-ARF, and ALW-J, respectively, as a function of bin near-surface RH in
NCP during the haze episode to further assess the ALW effect. The hourly near-surface RH in
$f_{base}$ is first divided into 28 bins, ranging from 30% to 100%, with interval of 2.5%. The
$PM_{2.5}$ contribution from the four sensitivity simulations is assembled in the same grid cell as
the bin RH, and an average of $PM_{2.5}$ contribution in each bin is calculated. The ALW does not
continuously increase near-surface [$PM_{2.5}$] with the RH. When the RH is less than 80%, the
near-surface $PM_{2.5}$ contribution of the ALW-TOT generally increases rapidly with the RH.
However, when the RH exceeds 80%, the contribution commences to decrease and fluctuates
between 20 and 30 µg m$^{-3}$, showing the effect of high occurrence frequencies of precipitation.
In addition, the ALW-HET dominates the $PM_{2.5}$ contribution, particularly with RH less than
50%. The ALW-RAD generally decreases [$PM_{2.5}$] slightly with the RH less than 52.5% and
vice versa with the RH more than 52.5%. The $PM_{2.5}$ contribution of the ALW-J is negative
and less than 1.5 µg m$^{-3}$, except when the RH is between 92.5% and 97.5%.






## 4   Conclusions and summaries


The good relationships between near-surface [PM$_{2.5}$] concentration and RH during the
wintertime of 2015 in NCP indicate the possibility that high RH plays an important role in the
PM$_{2.5}$ formation during the haze pollution. A severe haze pollution episode from 05
December 2015 to 04 January 2016 is simulated using the WRF-CHEM model to investigate
the impact of ALW caused by the accumulated moisture on near surface [PM$_{2.5}$] in NCP. The
air over NCP during the haze episode is humid, with an average simulated RH of about 71%.
In general, the WRF-CHEM model reproduces reasonably well the temporal variations of RH
compared to observations at three sites in NCP, although the model biases still exist due to
the uncertainties in simulated meteorological fields.
The FSA method is used to evaluate the contribution of the ALW effect on ARF,
photochemistry and heterogeneous reactions to the wintertime PM$_{2.5}$ concentration in NCP.
Model results show that the ALW substantially increases the daytime AOD with an average
contribution of 46% in NCP during the episode, enhancing the aerosol backward scattering,
and also augments the Reff from 0.31 μm to 0.48 μm, approaching the peak band of solar
radiation, favoring the aerosol forward scattering. The ALW does not significantly attenuate
the incoming solar radiation at the ground surface to enhance the ARF, and the average
near-surface PM$_{2.5}$ contribution of the ALW-ARF is about 1.1 μg m$^{-3}$ in NCP during the
episode. Therefore, the ALW-ARF is not an important factor for heavy haze formation and its
contribution relies on the relative importance of the ALW induced enhancement of aerosol
backward and forward scattering. The ALW generally decreases the photolysis rate at the
surface level due to enhanced backward aerosol scattering, but the favored forward scattering
still possibly accelerates the photolysis. Additionally, the ALW increases the photolysis rate
in the upper and above PBL. On average, the ALW decreases near-surface daytime O$_3$



concentrations by 0.20 μg m$^{-3}$ and [PM$_{2.5}$] by 0.87 μg m$^{-3}$, playing a minor role in mitigating
the haze pollution.
The ALW substantially enhances the SA formation by serving as an important medium
for liquid-phase and heterogeneous reactions. The ALW contribution to near-surface sulfate,
nitrate, ammonium, and SOA concentrations is 71%, 10%, 26%, and 48% in NCP during the
episode, respectively. However, the enhanced SA due to the ALW-HET enlarges particle sizes
to facilitate the coagulation, decreasing the aerosol number concentration and surface area
and further AOD. Therefore, the decreased AOD and increased Reff enhance the incoming
solar radiation reaching the ground surface, further increasing the surface temperature and
PBLH and decreasing near-surface primary aerosols. The ALW-HET contributes 15.9% of
near-surface [PM$_{2.5}$], less than the total SA enhancement of 18.4%, and the rest is
counterbalanced by the decrease in primary aerosols.
The near-surface PM$_{2.5}$ contribution of the ALW total effect is 17.5% in NCP, indicating
that ALW plays an important role in the PM$_{2.5}$ formation during the wintertime haze pollution.
Moreover, the ALW-HET overwhelmingly dominates the PM$_{2.5}$ enhancement due to the ALW.
The ALW does not consistently enhance near-surface [PM$_{2.5}$] with increasing RH. When the
RH exceeds 80%, the contribution of the ALW commences to decrease caused by the high
occurrence frequencies of precipitation.
Although the model performs reasonably well in simulating air pollutants, aerosol
species and optical properties, and RH during the episode in NCP, the uncertainties from
meteorological fields and emission inventory still exist, leading to model biases. In addition,
simulation period of one month might not be sufficient to provide a comprehensive view of
the ALW effect on the PM$_{2.5}$ formation. More wintertime case studies will need to be
performed in the future to further investigate the ALW effect.




*Author contribution.* Guohui Li, as the contact author, provided the ideas and financial support, developed the model code, verified the conclusions, and revised the paper. Jiarui Wu conducted a research, designed the experiments, carried the methodology out, performed the simulation, processed the data, prepared the data visualization, and prepared the manuscript with contributions from all authors. Naifang Bei provided the treatment of meteorological data, analyzed the study data, validated the model performance, and reviewed the manuscript. Bo Hu provided the observation data used in the study, synthesized the observation, and reviewed the paper. Suixin Liu, Meng Zhou, Qiyuan Wang, Zirui Liu, and Yichen Wang provided the data and the primary data process, and reviewed the manuscript. Xia Li, Lang Liu, and Tian Feng analyzed the initial simulation data, visualized the model results and reviewed the paper. Junji Cao, Xuexi Tie, Jun Wang provided critical reviews pre-publication stage. Luisa T. Molina provided a critical preview and financial support, and revised the manuscript.

*Acknowledgements.* This work is financially supported by the National Key R&D Plan (Quantitative Relationship and Regulation Principle between Regional Oxidation Capacity of Atmospheric and Air Quality (2017YFC0210000)) and National Research Program for Key Issues in Air Pollution Control. Luisa Molina acknowledges support from US NSF Award 1560494.





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

Y. F.: Sources of secondary organic aerosols in the Pearl River Delta region in fall:
Contributions from the aqueous reactive uptake of dicarbonyls, Atmos. Environ., 76,
200-207, 10.1016/j.atmosenv.2012.12.005, 2013.
Liggio, J., Li, S. M., and McLaren, R.: Reactive uptake of glyoxal by particulate matter, J.
Geophys. Res.-Atmos., 110, doi: 10.1029/2004jd005113, 2005.



Liu, Q., Jia, X. C., Quan, J. N., Li, J. Y., Li, X., Wu, Y. X., Chen, D., Wang, Z. F., and Liu, Y.
G.: New positive feedback mechanism between boundary layer meteorology and
secondary aerosol formation during severe haze events, Sci. Rep., 8, 8,
10.1038/s41598-018-24366-3, 2018.

Nguyen, T. B., Roach, P. J., Laskin, J., Laskin, A., and Nizkorodov, S. A.: Effect of humidity
on the composition of isoprene photooxidation secondary organic aerosol, Atmos. Chem.
Phys., 11, 6931-6944, 10.5194/acp-11-6931-2011, 2011.

Parrish, D. D. and Zhu, T.: Clean Air for Megacities, Science, 326, 674–675,
https://doi.org/10.1126/science.1176064, 2009.

Pathak, R. K., Wang, T., and Wu, W. S.: Nighttime enhancement of $PM_{2.5}$ nitrate in
ammonia-poor atmospheric conditions in Beijing and Shanghai: Plausible contributions
of heterogeneous hydrolysis of $N_2O_5$ and $HNO_3$ partitioning, Atmos. Environ., 45,
1183-1191, 10.1016/j.atmosenv.2010.09.003, 2011.

Penner, J. E., Hegg, D., and Leaitch, R.: Unraveling the role of aerosols in climate change,
Environ. Sci. Technol., 35, 332A-340A, 10.1021/es0124414, 2001.

Poulain, L., Wu, Z., Petters, M. D., Wex, H., Hallbauer, E., Wehner, B., Massling, A.,
Kreiden- weis, S. M., and Stratmann, F.: Towards closing the gap between hygroscopic
growth and CCN activation for secondary organic aerosols - Part 3: Influence of the
chemical compo- sition on the hygroscopic properties and volatile fractions of aerosols,
Atmos. Chem. Phys., 10, 3775–3785, doi:10.5194/acp-10-3775-2010, 2010.

Pilinis, C., Seinfeld, J. H., and Grosjean, D.: Water content of atmospheric aerosols, Atmos.
Environ., 23, 1601–1606, 1989.

Quan, J. N., Gao, Y., Zhang, Q., Tie, X. X., Cao, J. J., Han, S. Q., Meng, J. W., Chen, P. F.,
and Zhao, D. L.: Evolution of planetary boundary layer under different weather
conditions, and its impact on aerosol concentrations, Particuology, 11, 34-40,
10.1016/j.partic.2012.04.005, 2013.

Randles, C. A., Russell, L. M., and Ramaswamy, V.: Hygroscopic and optical properties of
organic sea salt aerosol and consequences for climate forcing, Geophys. Res. Lett., 31, 4,
10.1029/2004gl020628, 2004.

Riemer, N., Vogel, H., Vogel, B., Schell, B., Ackermann, I., Kessler, C., and Hass, H.: Impact
of the heterogeneous hydrolysis of $N_2O_5$ on chemistry and nitrate aerosol formation in
the lower troposphere under photosmog conditions, J. Geophys. Res.-Atmos., 108, 21,
10.1029/2002jd002436, 2003.

Seinfeld, J. H., Erdakos, G. B., Asher, W. E., and Pankow, J. F.: Modeling the formation of
secondary organic aerosol (SOA). 2. The predicted effects of relative humidity on
aerosol formation in the alpha-pinene-, beta-pinene-, sabinene-, Delta(3)-Carene-, and
cyclohexene-ozone systems, Environ. Sci. Technol., 35, 1806-1817, 10.1021/es001765+,
2001.

Seinfeld, J. H. and Pandis, S. N.: Atmospheric Chemistry and Physics: From Air Pollution to
Climate Change, John Wiley & Sons, USA, 1986.

Sun, Y., Wang, Z., Fu, P., Jiang, Q., Yang, T., Li, J., and Ge, X.: The impact of relative
humidity on aerosol composition and evolution processes during wintertime in Beijing,
China, Atmos. Environ., 77, 927-934, 10.1016/j.atmosenv.2013.06.019, 2013.



Stein, U., and Alpert, P.: Factor separation in numerical simulations, Journal of the
Atmospheric Science, 50, 2107-2115, 10.1175/1520-0469(1993)
050<2107:fsins>2.0.co;2, 1993.
Tao, J. C., Zhao, C. S., Ma, N., and Liu, P. F.: The impact of aerosol hygroscopic growth on
the single-scattering albedo and its application on the $NO_2$ photolysis rate coefficient,
Atmos. Chem. Phys., 14, 12055-12067, 10.5194/acp-14-12055-2014, 2014.
Tie, X. X., Huang, R. J., Cao, J. J., Zhang, Q., Cheng, Y. F., Su, H., Chang, D., Poschl, U.,
Hoffmann, T., Dusek, U., Li, G. H., Worsnop, D. R., and O'Dowd, C. D.: Severe
Pollution in China Amplified by Atmospheric Moisture, Sci. Rep., 7, 8,
10.1038/s41598-017-15909-1, 2017.
Volkamer, R., Martini, F. S., Molina, L. T., Salcedo, D., Jimenez, J. L., and Molina, M. J.: A
missing sink for gas-phase glyoxal in Mexico City: Formation of secondary organic
aerosol, Geophys. Res. Lett., 34, 5, 10.1029/2007gl030752, 2007.
Wang, G., Zhang, R., Gomez, M. E., Yang, L., Zamora, M. L., Hu, M., Lin, Y., Peng, J., Guo,
S., and Meng, J.: Persistent sulfate formation from London Fog to Chinese haze, P. Natl.
Acad. Sci. USA., 113, 13630–13635, 2016.
Wang, Y. S., Yao, L., Wang, L. L., Liu, Z. R., Ji, D. S., Tang, G. Q., Zhang, J. K., Sun, Y., Hu,
B., and Xin, J. Y.: Mechanism for the formation of the January 2013 heavy haze
pollution episode over central and eastern China, Sci. China-Earth Sci., 57, 14-25,
10.1007/s11430-013-4773-4, 2014.
Weinhold, B.: Ozone nation – EPA standard panned by the people, Environ. Health. Persp.,
116, A302–A305, 2008.
Wu, J. R., Li, G. H., Cao, J. J., Bei, N. F., Wang, Y. C., Feng, T., Huang, R. J., Liu, S. X.,
Zhang, Q., and Tie, X. X.: Contributions of trans-boundary transport to summertime air
quality in Beijing, China, Atmos. Chem. Phys., 17, 2035-2051,
10.5194/acp-17-2035-2017, 2017.
Wu, P., Ding, Y. H., and Liu, Y. J.: Atmospheric circulation and dynamic mechanism for
persistent haze events in the Beijing-Tianjin-Hebei region, Adv. Atmos. Sci., 34,
429-440, 10.1007/s00376-016-6158-z, 2017.
Wu, Z. J., Wang, Y., Tan, T. Y., Zhu, Y. S., Li, M. R., Shang, D. J., Wang, H. C., Lu, K. D.,
Guo, S., Zeng, L. M., and Zhang, Y. H.: Aerosol Liquid Water Driven by Anthropogenic
Inorganic Salts: Implying Its Key Role in Haze Formation over the North China Plain,
Environ. Sci. Technol. Lett., 5, 160-166, 10.1021/acs.estlett.8b00021, 2018.
Xing, L., Wu, J., Elser, M., Tong, S., Liu, S., Li, X., Liu, L., Cao, J., Zhou, J., El-Haddad, I.,
Huang, R., Ge, M., Tie, X., Prévôt, A. S. H., and Li, G.: Wintertime secondary organic
aerosol formation in Beijing-Tianjin-Hebei (BTH): Contributions of HONO sources and
heterogeneous reactions, Atmos. Chem. Phys. Discuss., 2018, 1-25,
10.5194/acp-2018-770, 2018.
Xu, J. W., Martin, R. V., van Donkelaar, A., Kim, J., Choi, M., Zhang, Q., Geng, G., Liu, Y.,
Ma, Z., Huang, L., Wang, Y., Chen, H., Che, H., Lin, P., and Lin, N.: Estimating
ground-level PM2.5 in eastern China using aerosol optical depth determined from the
GOCI satellite instrument, Atmos. Chem. Phys., 15, 13133-13144,
10.5194/acp-15-13133-2015, 2015.



Zhang, L., Sun, J. Y., Shen, X. J., Zhang, Y. M., Che, H., Ma, Q. L., Zhang, Y. W., Zhang, X.
Y., and Ogren, J. A.: Observations of relative humidity effects on aerosol light scattering
in the Yangtze River Delta of China, Atmos. Chem. Phys., 15, 8439-8454,
10.5194/acp-15-8439-2015, 2015.

Zhang, Q., Streets, D. G., Carmichael, G. R., He, K. B., Huo, H., Kannari, A., Klimont, Z.,
Park, I. S., Reddy, S., Fu, J. S., Chen, D., Duan, L., Lei, Y., Wang, L. T., and Yao, Z. L.:
Asian emissions in 2006 for the NASA INTEX-B mission, Atmos. Chem. Phys., 9,
5131-5153, https://doi.org/10.5194/acp-9-5131-2009, 2009.

Zhang, R. H., Li, Q., and Zhang, R. N.: Meteorological conditions for the persistent severe
fog and haze event over eastern China in January 2013, Sci. China-Earth Sci., 57, 26-35,
10.1007/s11430-013-4774-3, 2014.

Zhang, R., Jing, J., Tao, J., Hsu, S.-C., Wang, G., Cao, J., Lee, C. S. L., Zhu, L., Chen, Z.,
Zhao, Y., and Shen, Z.: Chemical characterization and source apportionment of $PM_{2.5}$ in
Beijing: seasonal perspective, Atmos. Chem. Phys., 13, 7053–7074,
https://doi.org/10.5194/acp-13-7053-2013, 2013.

Zhao, J., Levitt, N. P., Zhang, R. Y., and Chen, J. M.: Heterogeneous reactions of
methylglyoxal in acidic media: implications for secondary organic aerosol formation,
Environ. Sci. Technol., 40, 7682–7687, 2006.

Zhou, Y., Zhang, H. F., Parikh, H. M., Chen, E. H., Rattanavaraha, W., Rosen, E. P., Wang, W.
X., and Kamens, R. M.: Secondary organic aerosol formation from xylenes and mixtures
of toluene and xylenes in an atmospheric urban hydrocarbon mixture: Water and particle
seed effects (II), Atmos. Environ., 45, 3882-3890, 10.1016/j.atmosenv.2010.12.048,
2011.



**758**  Table 1 WRF-CHEM model configurations
**759**

| Regions | East Asia |
|---|---|
| Simulation period | December 05, 2015 - January 04, 2016 |
| Domain size | 400 × 400 |
| Domain center | 35°N, 114°E |
| Horizontal resolution | 12km × 12km |
| Vertical resolution | 35 vertical levels with a stretched vertical grid with spacing ranging from 30 m near the surface, to 500 m at 2.5 km and 1 km above 14 km |
| Microphysics scheme | WSM 6-class graupel scheme (Hong and Lim, 2006) |
| Boundary layer scheme | MYJ TKE scheme (Janjić, 2002) |
| Surface layer scheme | MYJ surface scheme (Janjić, 2002) |
| Land-surface scheme | Unified Noah land-surface model (Chen and Dudhia, 2001) |
| Longwave radiation scheme | Goddard longwave scheme (Chou and Suarez, 2001) |
| Shortwave radiation scheme | Goddard shortwave scheme (Chou and Suarez, 1999) |
| Meteorological boundary and initial conditions | NCEP 1°×1° reanalysis data |
| Chemical initial and boundary conditions | MOZART 6-hour output (Horowitz et al., 2003) |
| Anthropogenic emission inventory | Developed by Zhang et al. (2009) and Li et al. (2017), 2012 base year, and SAPRC-99 chemical mechanism |
| Biogenic emission inventory | MEGAN model developed by Guenther et al. (2006) |
| Model spin-up time | 28 hours |

**760**

**761**

**762**

**763**
**764**



**Figure Captions**


Figure 1 (a) WRF-CHEM simulation domain with topography and (b) North China Plain. In
(a), the blue circles represent centers of cities with ambient monitoring sites and the
size of circles denotes the number of ambient monitoring sites of cities. In (b), the red
capitals denote six typical polluted cities in NCP. A: Beijing; B: Tianjin; C:
Shijiazhuang; D: Baoding; E: Tangshan; F: Chengde. The blue numbers denote the
CERN sites with the hourly RH measurement. 1: Jiaozhouwan; 2: Yucheng; 3:
Luancheng.
Figure 2 Scatter plots of near-surface [$PM_{2.5}$] and RH at six typical polluted cities in NCP
during the 2015 wintertime. The red diamond shows the bin average of near-surface
[$PM_{2.5}$], and the red line denotes the variation of the bin average of near-surface
[$PM_{2.5}$] with RH.
Figure 3 Comparison of measured (black dots) and predicted (red line) diurnal profiles of the
RH in (a) Luancheng, (b) Yucheng, and (c) Jiaozhouwan from 05 December 2015 to
04 January 2016.
Figure 4 Spatial distribution of (a) NCEP reanalyzed and (b) simulated RH averaged from 05
December 2015 to 04 January 2016.
Figure 5 (a) absolute and (b) relative AOD contribution caused by the ALW, averaged from
05 December 2015 to 04 January 2016.
Figure 6 Near-surface [$PM_{2.5}$] contribution caused by the ALW-ARF, averaged from 05
December 2015 to 04 January 2016.
Figure 7 Average variations of AOD and Reff in $f_{base}$ (red line) and $f_{alw-rad0}$ (blue line)
as a function of bin [$PM_{2.5}$] in NCP during daytime from 05 December 2015 to 04
January 2016.
Figure 8 Average (a) percentage decrease of SWDOWN at the ground surface, (b) decrease of
TSFC, (c) percentage decrease of PBLH, and (d) percentage contribution of
near-surface [$PM_{2.5}$] caused by the ALW-ARF, as a function of the near-surface [$PM_{2.5}$]
in NCP during daytime from 05 December 2015 to 04 January 2016.
Figure 9 Average variations of daytime $NO_2$ photolysis and $O_3$ concentration at 1[st] and 5[th]
model layer (around 18 m and 420 m above the ground surface, respectively) caused
by ALW from 05 December 2015 to 04 January 2016 in NCP.
Figure 10 Near-surface $PM_{2.5}$ contribution caused by the ALW-J, averaged from 05
December 2015 to 04 January 2016 in NCP.
Figure 11 Near-surface sulfate, nitrate, and ammonium contribution caused by the ALW-HET,
averaged from 05 December 2015 to 04 January 2016.
Figure 12 Spatial distribution of $NH_3$ emission rate in December.
Figure 13 Near-surface SOA contribution caused by the ALW-HET, averaged from 05
December 2015 to 04 January 2016.
Figure 14 Near-surface $PM_{2.5}$ contribution caused by the ALW-HET, averaged from 05
December 2015 to 04 January 2016.
Figure 15 Average percentage decrease of (a) BC and (b) POA concentrations caused by the



ALW-HET, as a function of the near-surface $[PM_{2.5}]$ in NCP from 05 December 2015
to 04 January 2016.
Figure 16 Average variations of (a) AOD and (b) Reff in $\boldsymbol{f_{base}}$ (red line) and $\boldsymbol{f_{alw-het0}}$
(blue line), respectively, and average percentage decrease of near-surface (c) aerosol
number concentration and (d) surface area caused by the ALW-HET, as a function of
bin $[PM_{2.5}]$ in NCP from 05 December 2015 to 04 January 2016.
Figure 17 Average (a) percentage decrease of SWDOWN at the ground surface, (b) decrease
of TSFC, and (c) percentage decrease of PBLH caused by the ALW-HET, as a
function of the near-surface $[PM_{2.5}]$ in NCP during daytime from 05 December 2015
to 04 January 2016.
Figure 18 Near-surface $PM_{2.5}$ contribution caused by the ALW-TOT, averaged from 05
December 2015 to 04 January 2016 in NCP.
Figure 19 Average contributions to near-surface $[PM_{2.5}]$ caused by the ALW-TOT (black line),
ALW-HET (red line), ALW-RAD (green line), and ALW-J (blue line), respectively, as
a function of the RH in NCP from 05 December 2015 to 04 January 2016.



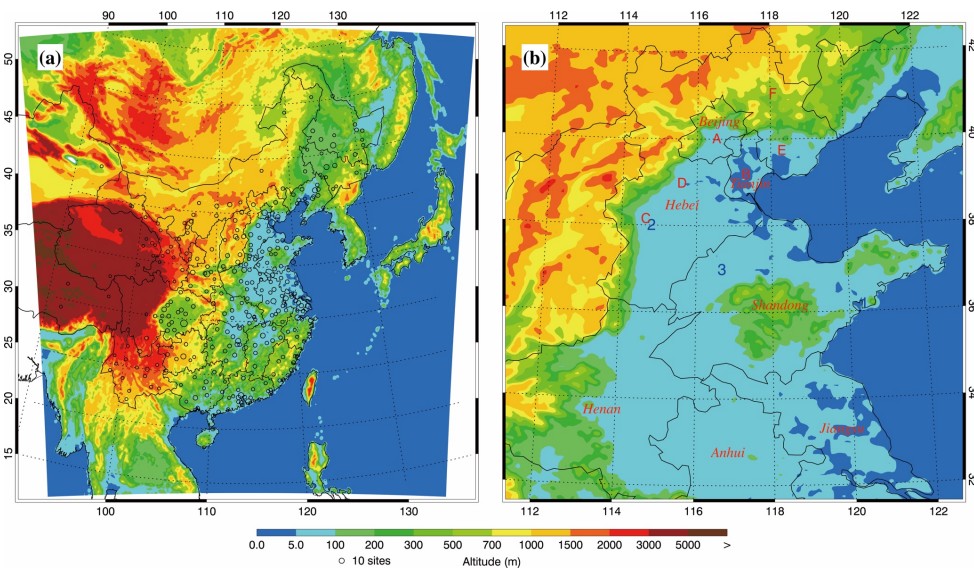

Figure 1 (a) WRF-CHEM simulation domain with topography and (b) North China Plain. In
(a), the blue circles represent centers of cities with ambient monitoring sites and the size of
blue circles denotes the number of ambient monitoring sites of cities. In (b), the red capitals
denote six typical polluted cities in NCP. A: Beijing; B: Tianjin; C: Shijiazhuang; D: Baoding;
E: Tangshan; F: Chengde. The blue numbers denote the CERN sites with the RH
measurement. 1: Jiaozhouwan; 2: Yucheng; 3: Luancheng.





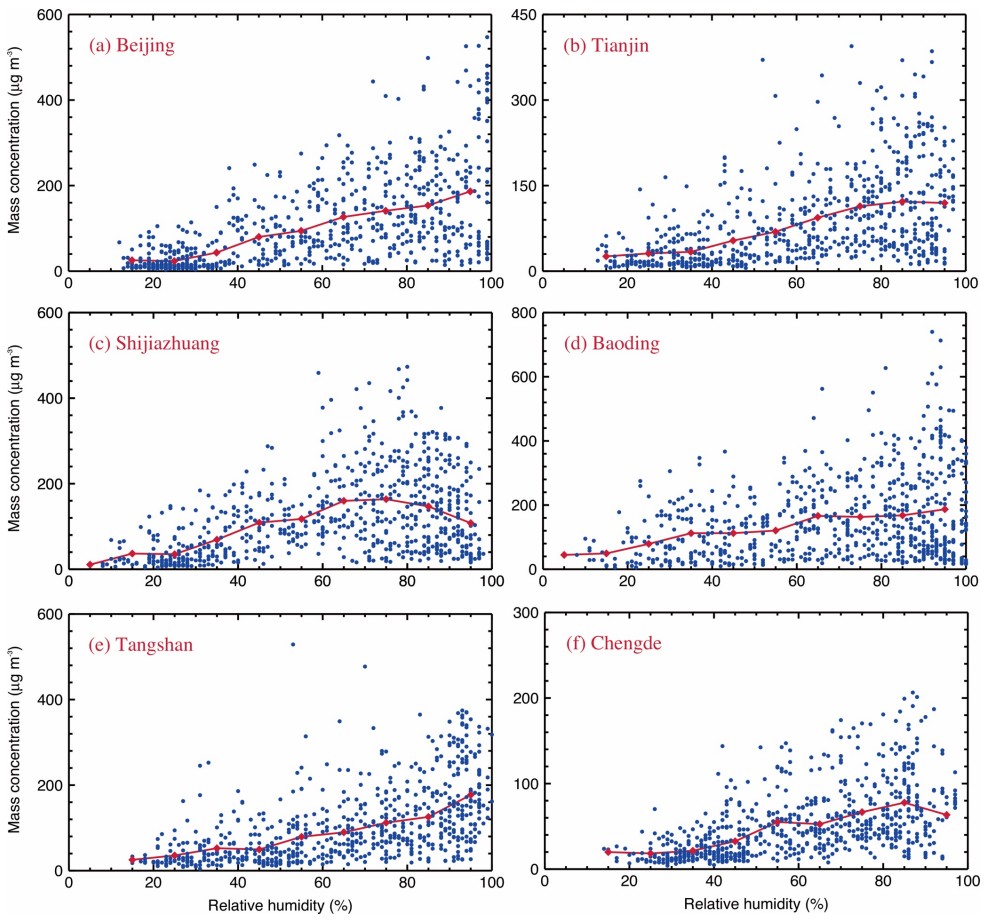

Figure 2 Scatter plots of near-surface [$PM_{2.5}$] and RH at six typical polluted cities in NCP
during the 2015 wintertime. The red diamond shows the bin average of near-surface [$PM_{2.5}$],
and the red line denotes the variation of the bin average of near-surface [$PM_{2.5}$] with RH.





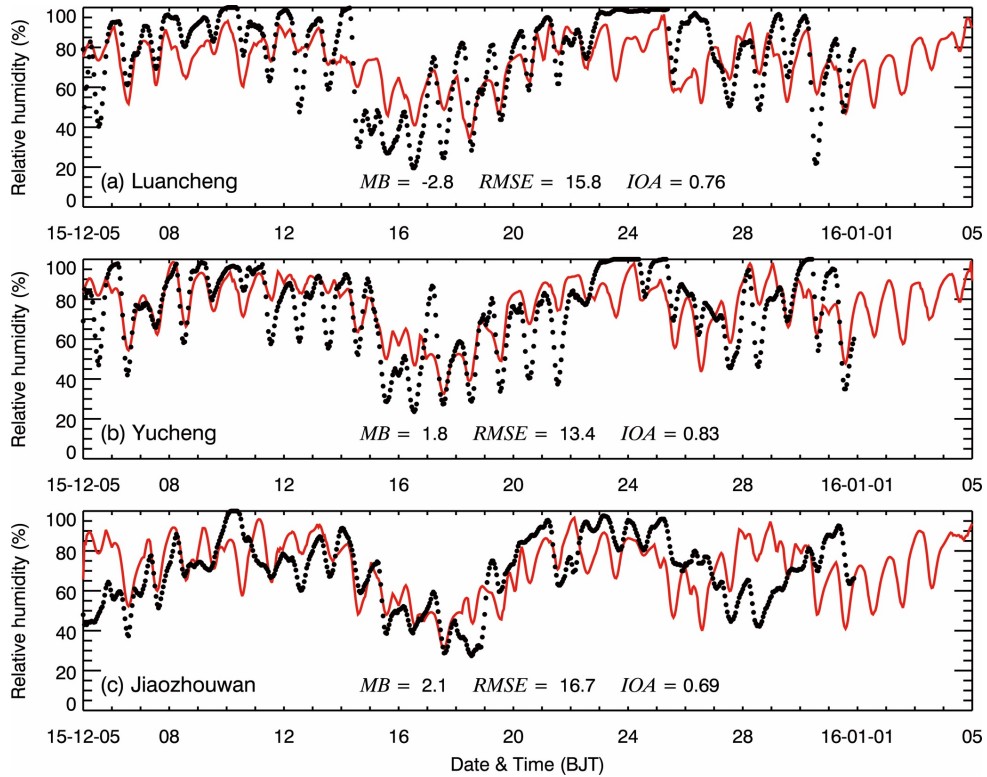

Figure 3 Comparison of measured (black dots) and predicted (red line) diurnal profiles of the
RH in (a) Luancheng, (b) Yucheng, and (c) Jiaozhouwan from 05 December 2015 to 04
January 2016.



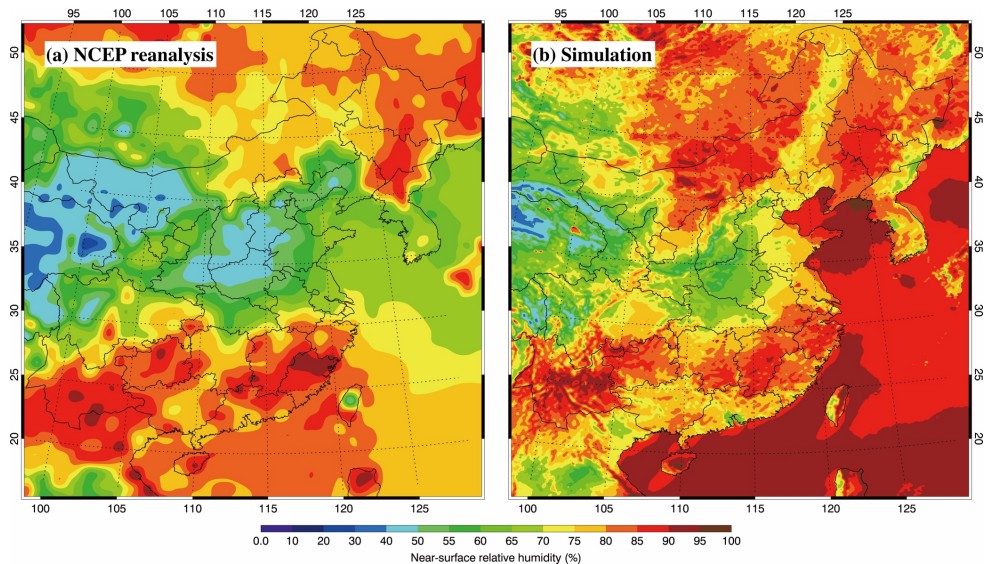

Figure 4 Spatial distribution of (a) NCEP reanalyzed and (b) simulated RH averaged from 05
December 2015 to 04 January 2016.



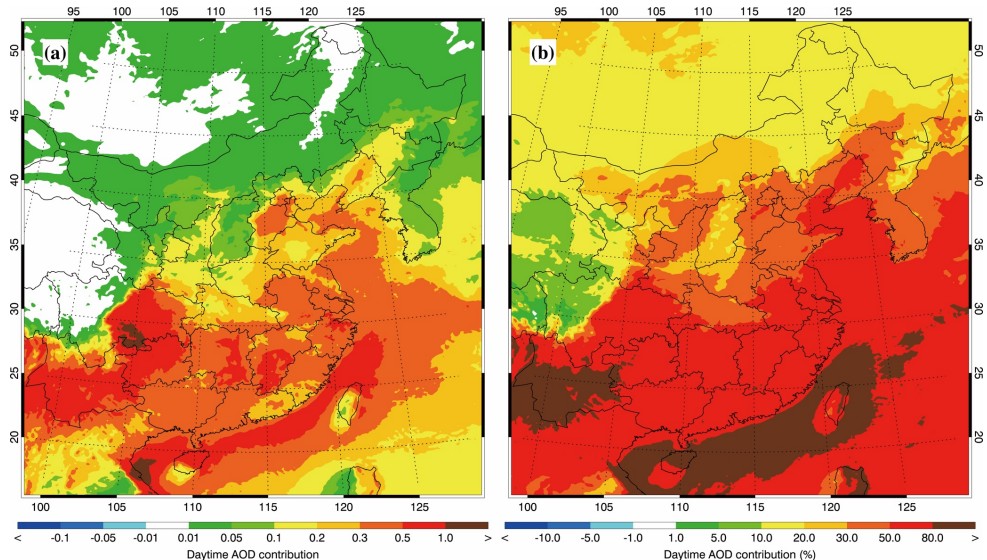


Figure 5 (a) absolute and (b) relative AOD contribution caused by the ALW, averaged from
05 December 2015 to 04 January 2016.





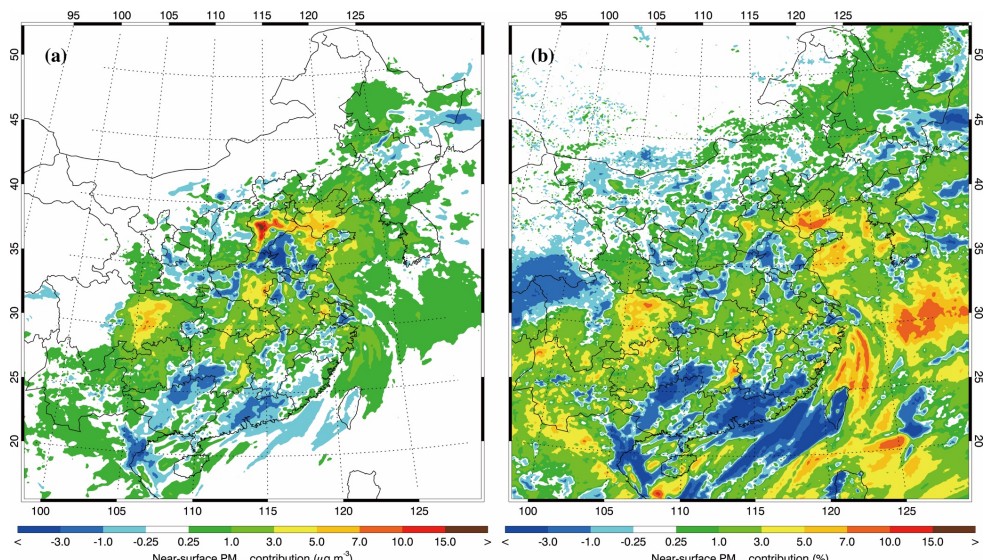

**877**

**878**

**879** Figure 6 Near-surface PM$_{2.5}$ contribution caused by the ALW-ARF, averaged from 05
**880** December 2015 to 04 January 2016 in NCP.

**881**

**882**

**883**

**884**

**885**





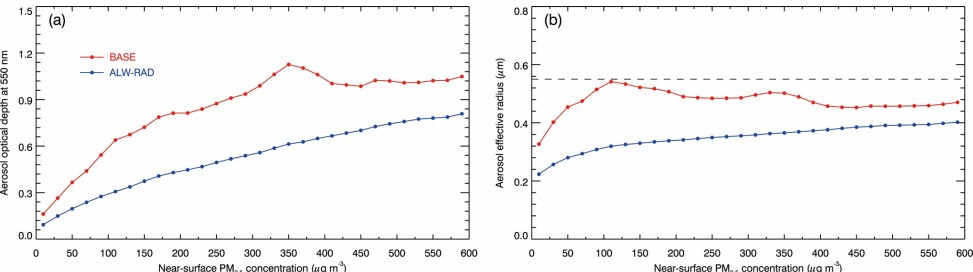

Figure 7 Average variations of AOD and Reff in $f_{base}$ (red line) and $f_{alw-rad0}$ (blue line)
as a function of bin [PM$_{2.5}$] in NCP during daytime from 05 December 2015 to 04 January
890   2016.





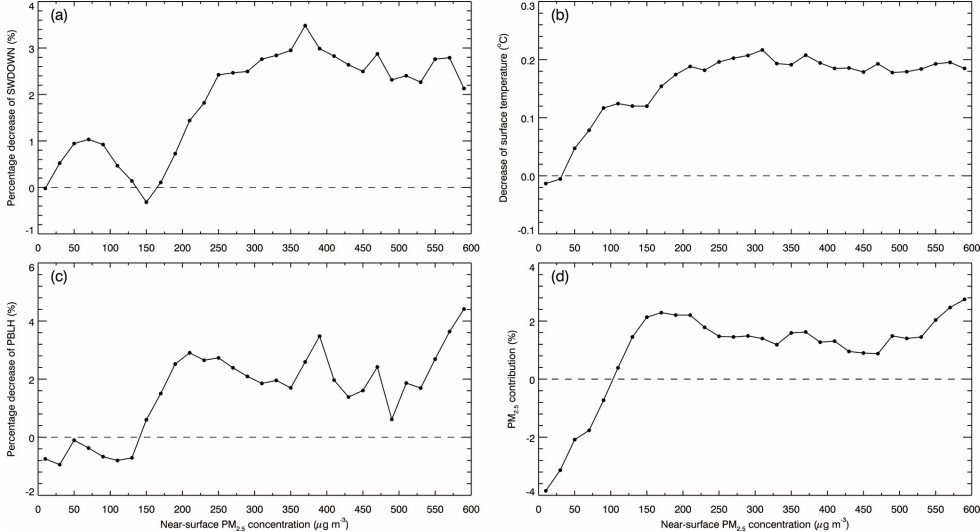

Figure 8 Average (a) percentage decrease of SWDOWN at the ground surface, (b) decrease of
TSFC, (c) percentage decrease of PBLH, and (d) percentage contribution of near-surface
[PM$_{2.5}$] caused by the ALW-ARF, as a function of the near-surface [PM$_{2.5}$] in NCP during
daytime from 05 December 2015 to 04 January 2016.





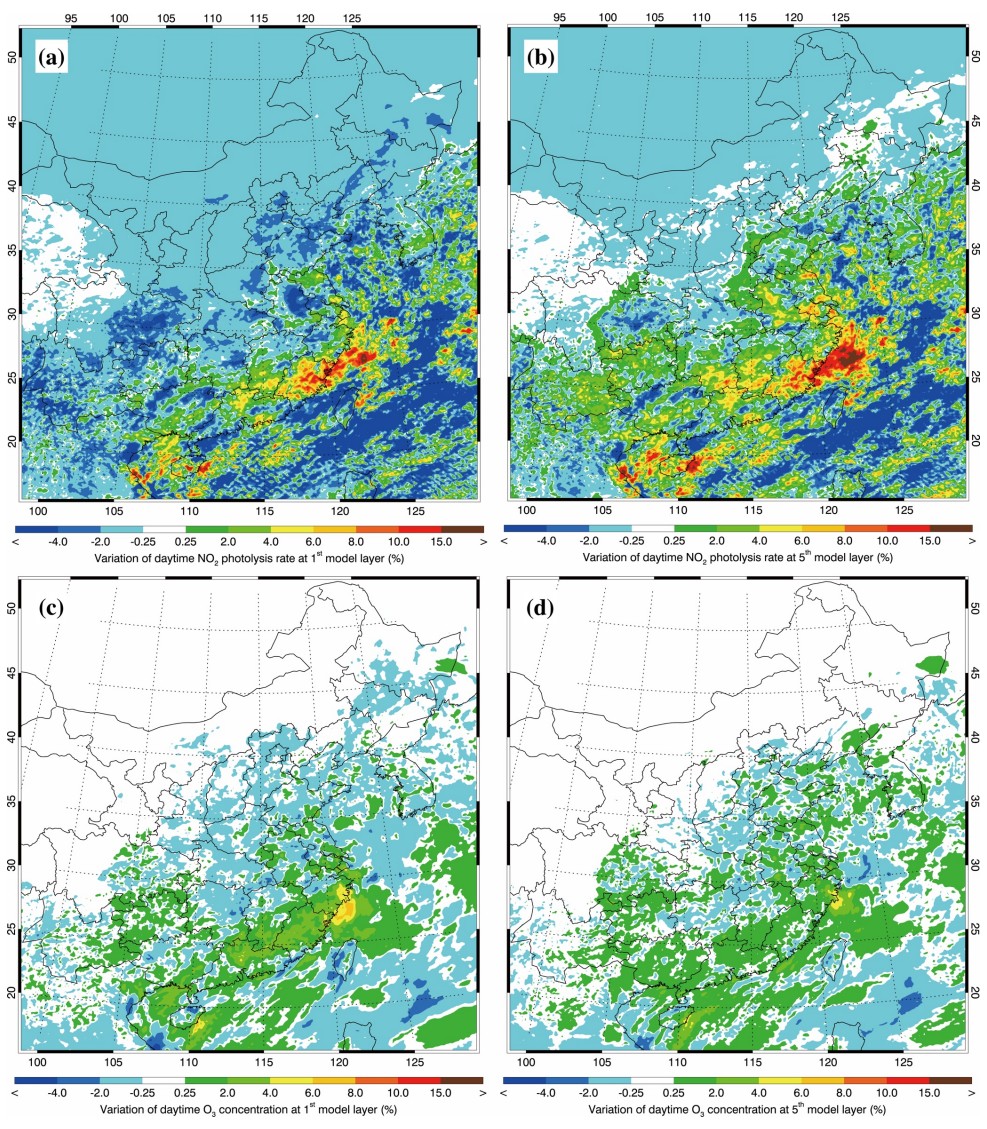

Figure 9 Average variations of daytime $NO_2$ photolysis and $O_3$ concentration at $1^{st}$ and $5^{th}$
model layer (around 18 m and 420 m above the ground surface, respectively) caused by ALW
from 05 December 2015 to 04 January 2016.





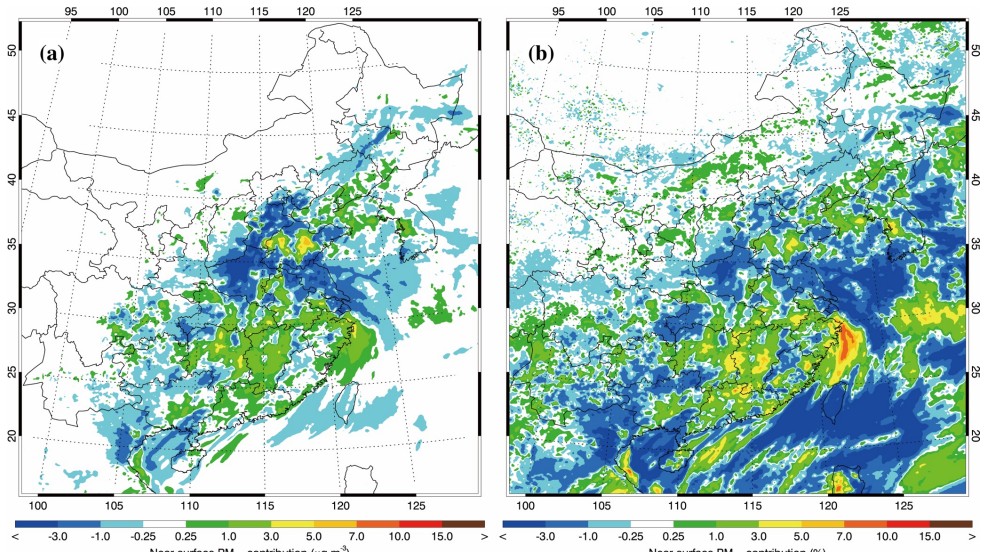

Figure 10 Near-surface PM$_{2.5}$ contribution caused by the ALW-J, averaged from 05
December 2015 to 04 January 2016.



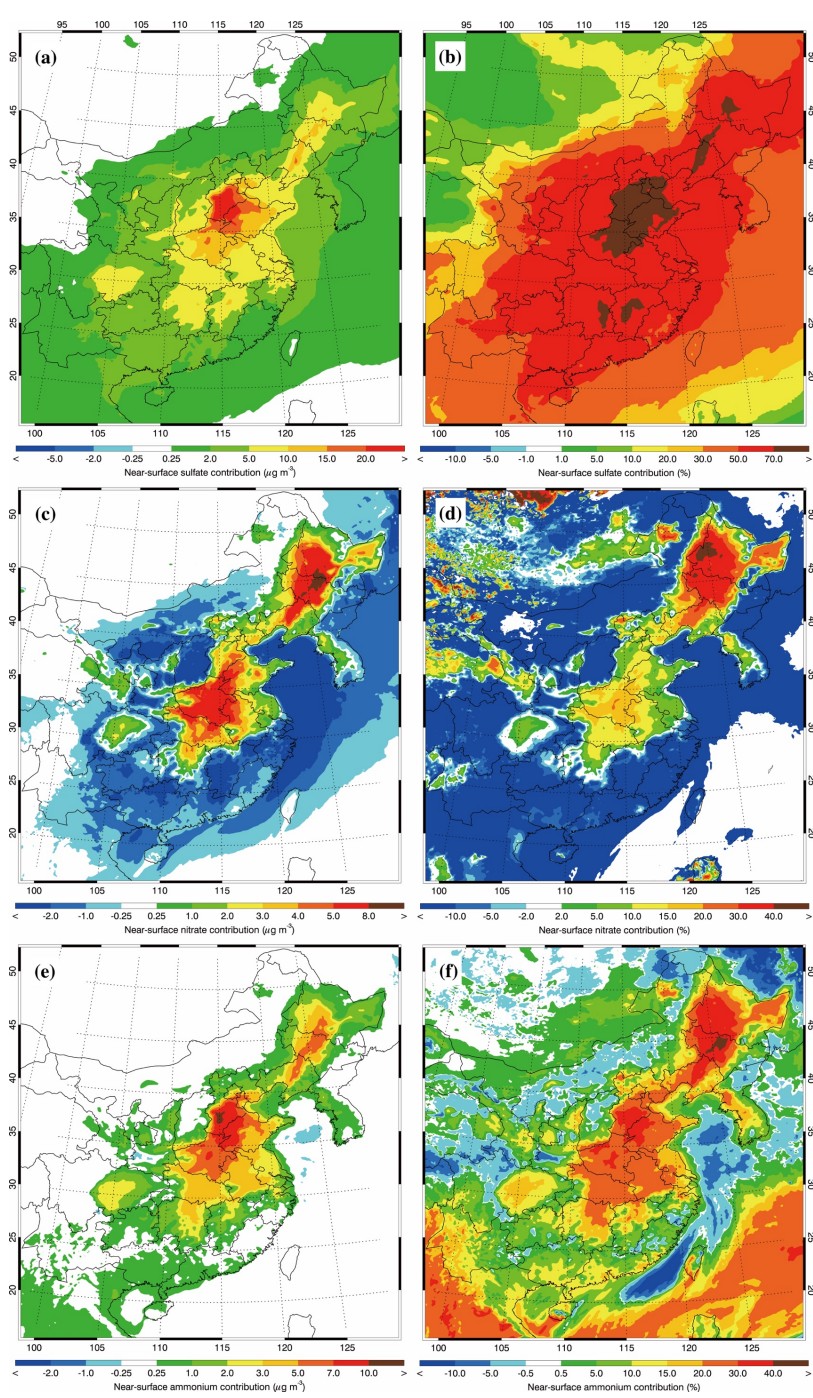

**926**

**927**

**928** Figure 11 Near-surface sulfate, nitrate, and ammonium contribution caused by the ALW-HET,

**929** averaged from 05 December 2015 to 04 January 2016.

**930**





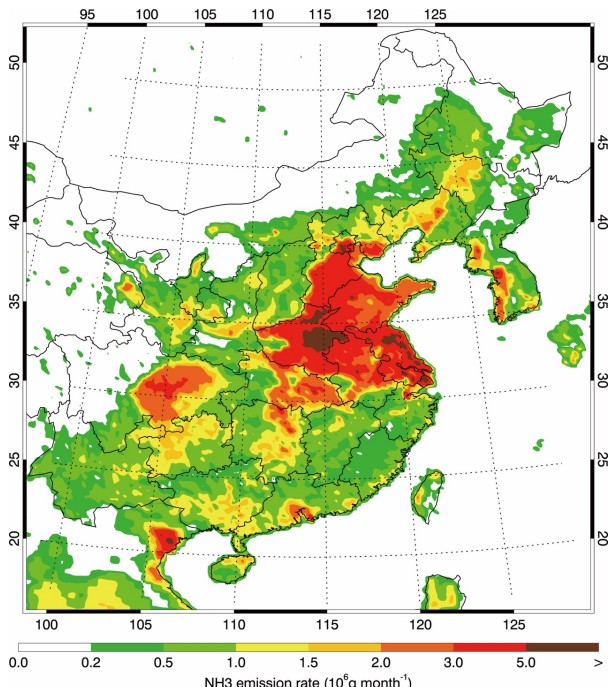



Figure 12 Spatial distribution of NH₃ emission rate in December.








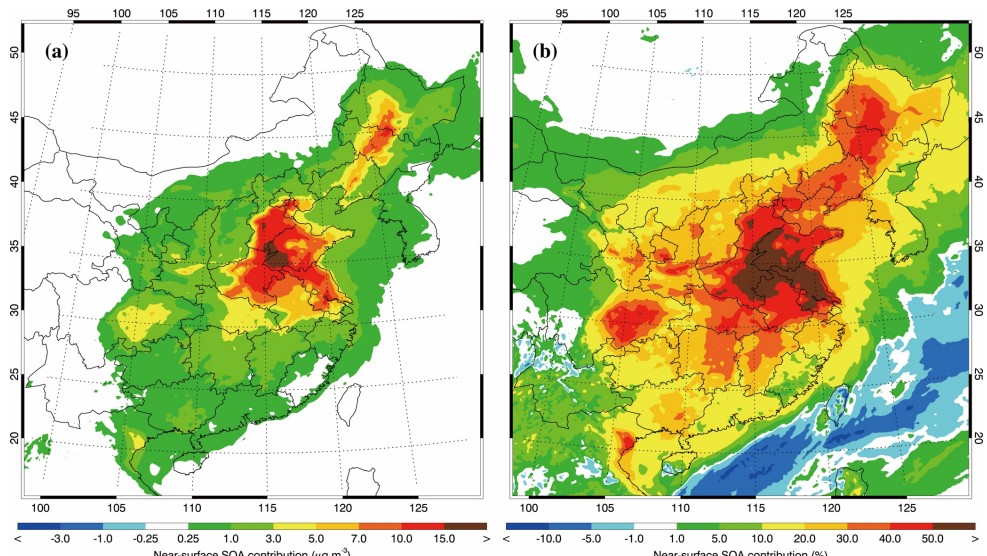

**939**

**940**

**941** Figure 13 Near-surface SOA contribution caused by the ALW-HET, averaged from 05

**942** December 2015 to 04 January 2016.

**943**

**944**

**945**

**946**

**947**





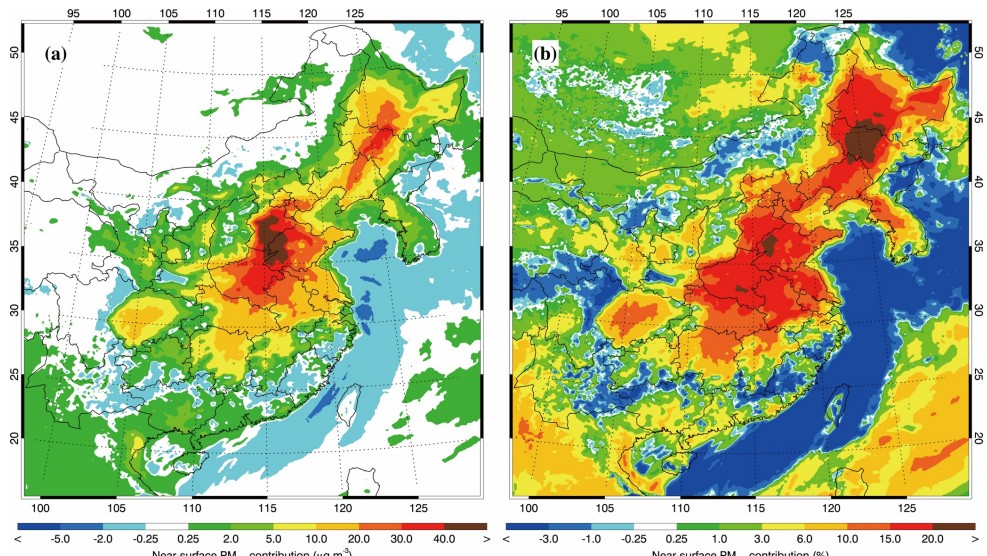

Figure 14 Near-surface [PM$_{2.5}$] contribution caused by the ALW-HET, averaged from 05
December 2015 to 04 January 2016.





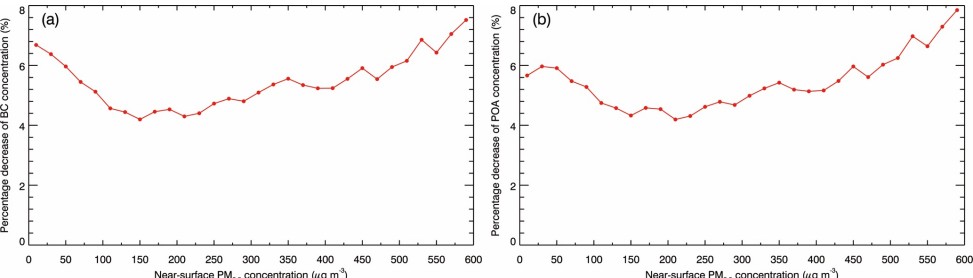

Figure 15 Average percentage decrease of (a) BC and (b) POA concentrations caused by the
ALW-HET, as a function of the near-surface [$PM_{2.5}$] in NCP from 05 December 2015 to 04
January 2016.





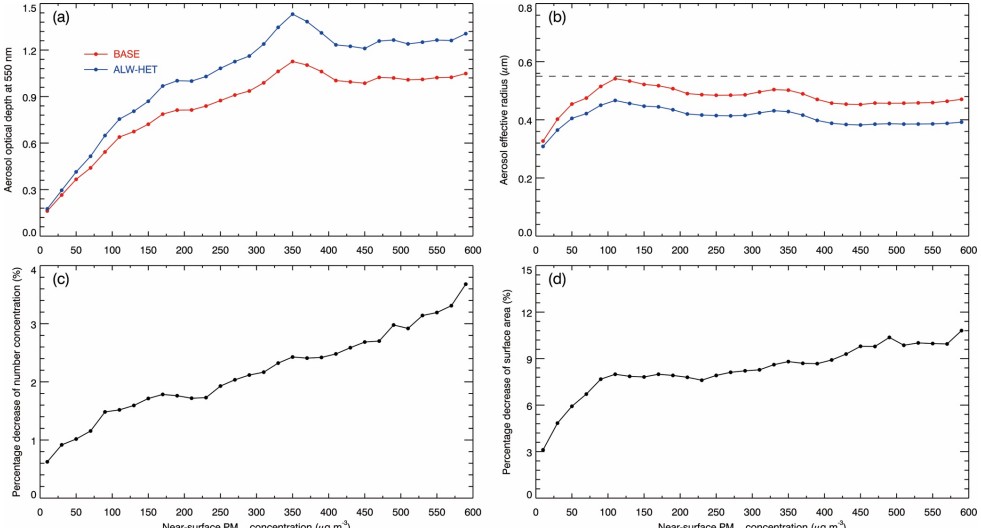

Figure 16 Average variations of (a) AOD and (b) Reff in $f_{base}$ (red line) and $f_{alw-het0}$ (blue line), respectively, and average percentage decrease of near-surface (c) aerosol number concentration and (d) surface area caused by the ALW-HET, as a function of bin [$PM_{2.5}$] in NCP from 05 December 2015 to 04 January 2016.

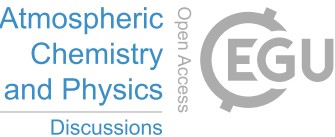

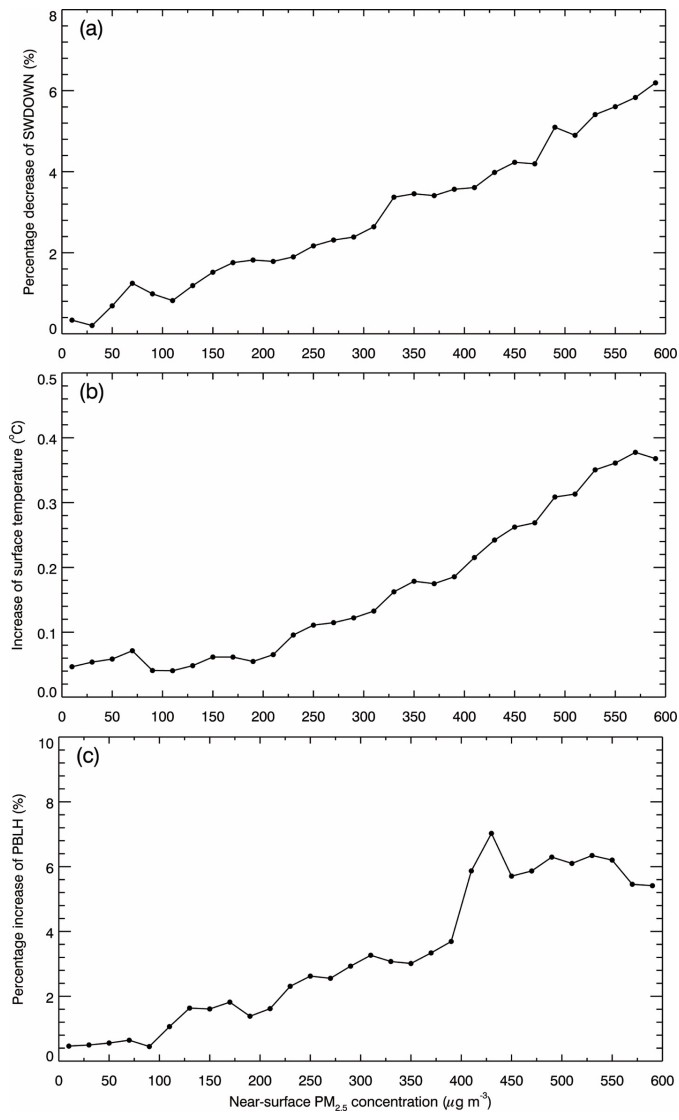

Figure 17 Average (a) percentage decrease of SWDOWN at the ground surface, (b) decrease
of TSFC, and (c) percentage decrease of PBLH caused by the ALW-HET, as a function of the
near-surface [$PM_{2.5}$] in NCP during daytime from 05 December 2015 to 04 January 2016.





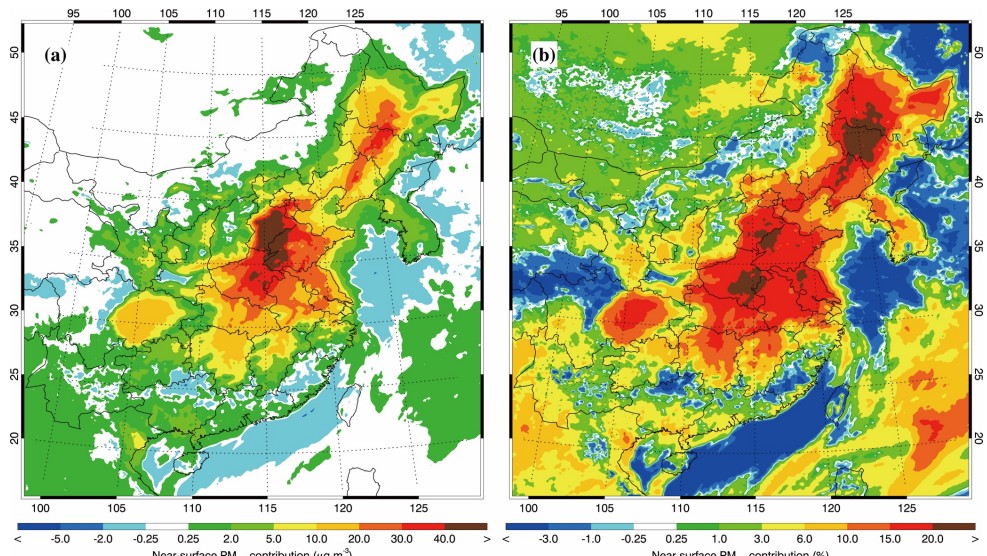

Figure 18 Near-surface PM$_{2.5}$ contribution caused by the ALW-TOT, averaged from 05 December 2015 to 04 January 2016.





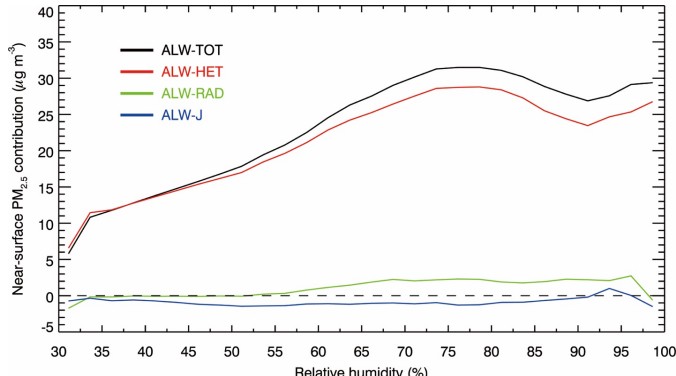

Figure 19 Average contributions to the near-surface [PM$_{2.5}$] caused by the ALW-TOT (black
line), ALW-HET (red line), ALW-RAD (green line), and ALW-J (blue line), respectively, as a
function of the RH in NCP from 05 December 2015 to 04 January 2016.