# Peer review of "Is water vapor a key player of the wintertime haze in North China Plain? 1 2 Jiarui Wu1,4,7, Naifang Bei2, Bo Hu3, Suixin Liu1,4, Meng Zhou5, Qiyuan Wang1,4, Xia Li1,4,7, Lang Liu1,4,7, Tian Feng1, Zir"

_Atmospheric Chemistry and Physics, 2018_

## Referee Comment (RC1) · Anonymous Referee #1 · 22 Mar 2019

The manuscript is a thorough modeling analysis of the role of aerosol liquid water (ALW) on radiation, photolysis, and heterogeneous chemistry and how these individual effects can feed back onto surface PM2.5 concentrations in North China Plain in winter. The strength of the paper is that it estimates the contribution of each factor separately in a modeling framework and thus leads to quantitative understanding of the importance of ALW on winter haze in China. My following comments ask the authors to provide more details on the modeling experiments as these are the foundations of this work.

Line 208-212: Describe in more details how the sensitivity simulations were performed. For each simulation, which are the specific mechanisms involving ALW that were turned off and how? For example, for the ARF simulation, one can turn off the hygroscopic growth of aerosols or alternatively not count the ALW in the AOD calculation. Which way did the authors take? For the role of aqueous/heterogeneous reactions, I presume the authors turned off these reactions in the model, but did the authors allow for the hygroscopic growth of aerosols? As the subsequent analysis and discussion rely on these sensitivity simulations, how they were performed in the WRF-Chem model should be described in sufficient details. Furthermore, the estimate of these effects is likely model dependent which further warrants a good description on the modeling experiments.

Line 213 and other places where f terms are used: Explain what $f_{alw}$ is. Is it the same as $f_{base}$? I would think so. Then the notation could be made more intuitively understandable to change it to $f_{base}$ throughout. For example, $f_{alw-rad0}$ should be $f_{base-rad0}$, then the difference between $f_{base}$ and $f_{base-rad0}$ represent the radiative effect.

About the figures: many figures have multiple panels but almost all the captions do not provide a clear description what each panel shows; for examples, Figures 6, 7, 9, 10, 11, 13, 14, 18, and 19. The standard practice is to include panel numbers (a, b, c, d, etc.) in the figure caption next to the description of the data shown in each panel.

---

## Referee Comment (RC2) · Anonymous Referee #3 · 15 Apr 2019

The paper used several sensitivity simulations to evaluate the impact of aerosol liquid water on wintertime haze in China. There are some major problems that the authors need to address before it can be considered for publication.

1) The authors did not provide a detailed description of how four sensitivity simulations are conducted, and how the contributions of aerosol water are determined. The methods section is too simple for meaningful evaluation of the results of the paper.

2) Aerosol liquid water content and aerosol composition are mutually influenced. I am concerned that simply comparing base case results with some sensitivity simulations without the aerosol water effect cannot untangle this.

Consider a simple ideal scenario in which particles in the air are pure dry NH4NO3 and

[Figure]

RH increases from 30% to just above the DRH of NH4NO3 (and remains above DRH). When this happens, the particle takes up water. This water can serve as a reaction medium or surface to facilitate the additional formation of secondary aerosols, such as (NH4)2SO4 and NH4NO3 (via aqueous reactions and/or heterogeneous reactions). These additional salt components lead to a further increase of aerosol liquid water content, which provides more reaction volume or surface to form even more aerosol salt components.

Now, how do we quantify the contributions of aerosol liquid water on the formation of SA? If I understand the author's method correctly, it will be determined from a base simulation with all processes and a sensitivity simulation without reactions in the aerosol water. However, this is logically incorrect because the initial water that triggers the subsequent reactions is due to the initial NH4NO3. If I replace NH4NO3 with hydrophobic BC, no water uptake will happen and no SA will form. Could I thus claim that all water and additional SA formed in the base case is due to the initial NH4NO3 seed particles?

The authors need to clearly describe how they solve this chicken-or-egg problem in attributing some effects to aerosol liquid water while the aerosol liquid water content also depends on the composition of the seed particles.

---

## Author Comment (AC1) · 26 May 2019

The comment was uploaded in the form of a supplement:
https://www.atmos-chem-phys-discuss.net/acp-2018-1289/acp-2018-1289-AC1-supplement.zip

―――――――――――――――――――――――――

---

## Author Comment (AC2) · 26 May 2019

The comment was uploaded in the form of a supplement:
https://www.atmos-chem-phys-discuss.net/acp-2018-1289/acp-2018-1289-AC2-
supplement.zip
* * *

---

## Author Comment (AC3) · 26 May 2019

The comment was uploaded in the form of a supplement:
https://www.atmos-chem-phys-discuss.net/acp-2018-1289/acp-2018-1289-AC3-supplement.zip
* * *

---

## Author Response (AR1)

May 26, 2019

Dear Editor,

We have received the comments from the two reviewers of the manuscript. Below are our responses and the revisions that we have made in the manuscript.

Thank you for your efforts on this manuscript. We look forward to hearing from you.

Best Regards,

Guohui Li

**Reply to Anonymous Referee #1**

We thank the reviewer for the careful reading of the manuscript and helpful comments. We have revised the manuscript following the suggestion, as described below.

The manuscript is a thorough modeling analysis of the role of aerosol liquid water (ALW) on radiation, photolysis, and heterogeneous chemistry and how these individual effects can feed back onto surface $PM_{2.5}$ concentrations in North China Plain in winter. The strength of the paper is that it estimates the contribution of each factor separately in a modeling framework and thus leads to quantitative understanding of the importance of ALW on winter haze in China. My following comments ask the authors to provide more details on the modeling experiments as these are the foundations of this work.

**1 Comment**: Line 208-212: Describe in more details how the sensitivity simulations were performed. For each simulation, which are the specific mechanisms involving ALW that were turned off and how? For example, for the ARF simulation, one can turn off the hygroscopic growth of aerosols or alternatively not count the ALW in the AOD calculation. Which way did the authors take? For the role of aqueous/heterogeneous reactions, I presume the authors turned off these reactions in the model, but did the authors allow for the hygroscopic growth of aerosols? As the subsequent analysis and discussion rely on these sensitivity simulations, how they were performed in the WRF-Chem model should be described in sufficient details. Furthermore, the estimate of these effects is likely model dependent which further warrants a good description on the modeling experiments.

**Response**: We have included a table (Table 2) to describe the sensitivity simulations in more details, and also clarified in Section 3.3: "*It is worth noting that in all the sensitivity simulations, the aerosol hygroscopic growth is not turned off. In the sensitivity simulation $f_{alw\_rad0}$, only the ALW contribution to the AOD is not included in the ARF. In the $f_{alw\_j0}$, only the ALW contribution to the AOD is not included in the photolysis calculation. In the $f_{alw\_het0}$, only the heterogeneous formation of secondary aerosols (SA) involving the ALW is turned off, including the $SO_2$ heterogeneous oxidation by $O_2$ catalyzed by $Fe^{3+}$, $N_2O_5$ heterogeneous hydrolysis, and the heterogeneous reaction of glyoxal and methylglyoxal. For the $f_{alw\_tot0}$, the ALW contribution to the AOD is not considered in the ARF and photolysis*

*calculation, and the SA heterogeneous formation involving the ALW is excluded. Detailed description about the sensitivity simulations can be found in Table 2.*"

Table 2 Description of the sensitivity simulations.

| Case | Aerosol hygroscopic growth | ALW contribution to AOD | | Multiphase reactions involving the ALW |
|------|----------------------------|-------------------------|---|----------------------------------------|
| | | in the ARF | in the photolysis | |
| $f_{base}$ | On | On | On | On |
| $f_{alw\_rad0}$ | On | Off | On | On |
| $f_{alw\_j0}$ | On | On | Off | On |
| $f_{alw\_het0}$ | On | On | On | Off |
| $f_{alw\_tot0}$ | On | Off | Off | Off |

**2 Comment:** Line 213 and other places where $f$ terms are used: Explain what $f_{alw}$ is. Is it the same as $f_{base}$? I would think so. Then the notation could be made more intuitively understandable to change it to $f_{base}$ throughout. For example, $f_{alw-rad0}$ should be $f_{base-rad0}$, then the difference between $f_{base}$ and $f_{base-rad0}$ represent the radiative effect.

**Response**: We have clarified in Section 3.3: "*Besides the base case with all the ALW effect (hereafter referred to as $f_{base}$), additional four sensitivity simulations have been performed, in which the ALW effect on the ARF, photolysis, multiphase reactions, and the total is excluded, respectively (hereafter referred to as $f_{alw\_rad0}$, $f_{alw\_j0}$, $f_{alw\_het0}$, and $f_{alw\_tot0}$, respectively).*", and "*The difference between $f_{base}$ and $f_{alw\_rad0}$ represents the ALW effect on the ARF during the study episode, so does for the ALW effect on photochemistry, multiphase reactions, and the total ALW effect.*". We have changed "$f_{alw-rad0}$, $f_{alw-j0}$, $f_{alw-het0}$, and $f_{alw-tot0}$" to "$f_{alw\_rad0}$, $f_{alw\_j0}$, $f_{alw\_het0}$, and $f_{alw\_tot0}$" in the manuscript.

**3 Comment:** About the figures: many figures have multiple panels but almost all the captions do not provide a clear description what each panel shows; for examples, Figures 6, 7, 9, 10, 11, 13, 14, 18, and 19. The standard practice is to include panel numbers (a, b, c, d, etc.) in the figure caption next to the description of the data shown in each panel.

**Response:** We have revised the figure caption of Figures 6, 7, 9, 10, 11, 13, 14, and 18 as suggested, and Figure 19 has no panel numbers.

*Figure 6 Average near-surface (a) absolute and (b) relative PM$_{2.5}$ contribution caused by the ALW-ARF from 05 December 2015 to 04 January 2016.*

*Figure 7 Average variations of (a) AOD and (b) Reff in $f_{base}$ (red line) and $f_{alw\_rad0}$ (blue line) as a function of bin [PM$_{2.5}$] in NCP during daytime from 05 December 2015 to 04 January 2016.*

*Figure 9 Average variations of daytime (a)/(b) NO$_2$ photolysis and (c)/(d) O$_3$ concentration at 1$^{st}$/5$^{th}$ model layer (around 18 m and 420 m above the ground surface, respectively) caused by the ALW from 05 December 2015 to 04 January 2016 in NCP.*

*Figure 10 Average near-surface (a) absolute and (b) relative PM$_{2.5}$ contribution caused by the ALW-J from 05 December 2015 to 04 January 2016.*

*Figure 11 Average near-surface (a)/(c)/(e) absolute and (b)/(d)/(f) relative contribution to sulfate/nitrate/ammonium concentrations, caused by the ALW-HET from 05 December 2015 to 04 January 2016.*

*Figure 13 Average near-surface (a) absolute and (b) relative SOA contribution caused by the ALW-HET from 05 December 2015 to 04 January 2016.*

*Figure 14 Average near-surface (a) absolute and (b) relative PM$_{2.5}$ contribution caused by the ALW-HET from 05 December 2015 to 04 January 2016.*

*Figure 18 Average near-surface (a) absolute and (b) relative PM$_{2.5}$ contribution caused by the ALW-TOT from 05 December 2015 to 04 January 2016.*

**Reply to Anonymous Referee #2**

We thank the reviewer for the careful reading of the manuscript and helpful comments. We have revised the manuscript following the suggestion, as described below.

The paper used several sensitivity simulations to evaluate the impact of aerosol liquid water on wintertime haze in China. There are some major problems that the authors need to address before it can be considered for publication.

**1 Comment**: The authors did not provide a detailed description of how four sensitivity simulations are conducted, and how the contributions of aerosol water are determined. The methods section is too simple for meaningful evaluation of the results of the paper.

**Response:** We have included a table (Table 2) to describe the sensitivity simulations in more details, and also clarified in Section 3.3: "*Besides the base case with all the ALW effect (hereafter referred to as $f_{base}$), additional four sensitivity simulations have been performed, in which the ALW effect on the ARF, photolysis, multiphase reactions, and the total is excluded, respectively (hereafter referred to as $f_{alw\_rad0}$, $f_{alw\_j0}$, $f_{alw\_het0}$, and $f_{alw\_tot0}$, respectively). It is worth noting that in all the sensitivity simulations, the aerosol hygroscopic growth is not turned off. In the sensitivity simulation $f_{alw\_rad0}$, only the ALW contribution to the AOD is not included in the ARF. In the $f_{alw\_j0}$, only the ALW contribution to the AOD is not considered in the photolysis calculation. In the $f_{alw\_het0}$, only the heterogeneous formation of secondary aerosols (SA) involving the ALW is turned off, including the $SO_2$ heterogeneous oxidation by $O_2$ catalyzed by $Fe^{3+}$, $N_2O_5$ heterogeneous hydrolysis, and the heterogeneous reaction of glyoxal and methylglyoxal. For the $f_{alw\_tot0}$, the ALW contribution to the AOD is not considered in the ARF and photolysis calculation, and the SA heterogeneous formation involving the ALW is excluded. Detailed description about the sensitivity simulations can be found in Table 2. The difference between $f_{base}$ and $f_{alw\_rad0}$ represents the ALW effect on the ARF during the study episode, and so does for the ALW effect on photochemistry, multiphase reactions, and the total ALW effect.*".

Table 2 Description of the sensitivity simulations.

| Case | Aerosol hygroscopic growth | ALW contribution to AOD | | Multiphase reactions involving the ALW |
| | | in the ARF | in the photolysis | |
|---|---|---|---|---|
| $f_{base}$ | On | On | On | On |
| $f_{alw-rad0}$ | On | Off | On | On |
| $f_{alw-j0}$ | On | On | Off | On |
| $f_{alw-het0}$ | On | On | On | Off |
| $f_{alw-tot0}$ | On | Off | Off | Off |

**2 Comment:** Aerosol liquid water content and aerosol composition are mutually influenced. I am concerned that simply comparing base case results with some sensitivity simulations without the aerosol water effect cannot untangle this.

Consider a simple ideal scenario in which particles in the air are pure dry $NH_4NO_3$ and RH increases from 30% to just above the DRH of $NH_4NO_3$ (and remains above DRH). When this happens, the particle takes up water. This water can serve as a reaction medium or surface to facilitate the additional formation of secondary aerosols, such as $(NH_4)_2SO_4$ and $NH_4NO_3$ (via aqueous reactions and/or heterogeneous reactions). These additional salt components lead to a further increase of aerosol liquid water content, which provides more reaction volume or surface to form even more aerosol salt components.

Now, how do we quantify the contributions of aerosol liquid water on the formation of SA? If I understand the author's method correctly, it will be determined from a base simulation with all processes and a sensitivity simulation without reactions in the aerosol water. However, this is logically incorrect because the initial water that triggers the subsequent reactions is due to the initial $NH_4NO_3$. If I replace $NH_4NO_3$ with hydrophobic BC, no water uptake will happen and no SA will form. Could I thus claim that all water and additional SA formed in the base case is due to the initial $NH_4NO_3$ seed particles?

The authors need to clearly describe how they solve this chicken-or-egg problem in attributing some effects to aerosol liquid water while the aerosol liquid water content also depends on the composition of the seed particles.

**Response:** The question is very interesting, although it is not the case in the atmosphere. Apparently, the ALW content (ALWC) and aerosol composition are mutually influenced, i.e., the ALWC depends on the existence of hydroscopic aerosols (mainly inorganic components)

and the relative humidity in the atmosphere, and the secondary aerosol (SA), particularly inorganic components, formed via heterogeneous and aqueous reactions involving the ALW further increases the ALWC. Considering an ideal atmosphere with only hydrophobic aerosols, the ALW effect will not happen because no water uptake occurs. Once hydroscopic aerosols exist in the ideal atmosphere, the ALW effect will be triggered to promote the SA formation under high RH conditions. It is certain that the ALW and additional SA formed in the ideal atmosphere is caused by the initial hydroscopic aerosol. However, in the real atmosphere, hydroscopic aerosols can be formed through the nucleation and condensation, distributed via the thermodynamic process, and directly emitted from various anthropogenic and natural sources. Therefore, in the model simulations, the initial hydroscopic aerosols or the seed particles, are insignificant to the ALWC because other processes dominate the hydroscopic aerosol concentration in the atmosphere, including direct emissions, nucleation, condensation et al., particularly after model spin-up. We have clarified in Section 3.3.3: "*It is worth noting that the ALW content (ALWC) and aerosol composition are mutually influenced, i.e., the ALWC depends on the existence of hydroscopic aerosols (mainly inorganic components) and the RH in the atmosphere, and the SA, particularly inorganic components, formed via heterogeneous and aqueous reactions involving the ALW further increases the ALWC. Hence the initial hydroscopic aerosols might play a seeding role in the contribution of the ALW-HET, likely constituting the most important factor in determining the ALWC and additional SA formed via heterogeneous or aqueous reactions. However, in model simulations, the initial hydroscopic aerosols or the seed particles, are insignificant to the ALWC, since even without consideration of multiphase formation, the other processes still dominate the hydroscopic aerosol concentration in the atmosphere, including direct emissions, nucleation, condensation et al., particularly after model spin-up. Therefore, the effect of initial hydroscopic aerosols or seed particles on the ALW and additional heterogeneous SA is generally negligible.*"